# REASONING WITH LATENT DIFFUSION IN OFFLINE REINFORCEMENT LEARNING

**Siddarth Venkatraman**[1,*] **Shivesh Khaitan**[2,*] **Ravi Tej Akella**[2,*] **John Dolan**[2]
**Jeff Schneider**[2] **Glen Berseth**[1]

[1]Mila, Université de Montréal    [2]Carnegie Mellon University    [*]Equal Contribution

`siddarth.venkatraman@mila.quebec`

## ABSTRACT

Offline reinforcement learning (RL) holds promise as a means to learn high-reward policies from a static dataset, without the need for further environment interactions. However, a key challenge in offline RL lies in effectively stitching portions of suboptimal trajectories from the static dataset while avoiding extrapolation errors arising due to a lack of support in the dataset. Existing approaches use conservative methods that are tricky to tune and struggle with multi-modal data (as we show) or rely on noisy Monte Carlo return-to-go samples for reward conditioning. In this work, we propose a novel approach that leverages the expressiveness of latent diffusion to model in-support trajectory sequences as compressed latent skills. This facilitates learning a Q-function while avoiding extrapolation error via batch-constraining. The latent space is also expressive and gracefully copes with multi-modal data. We show that the learned temporally-abstract latent space encodes richer task-specific information for offline RL tasks as compared to raw state-actions. This improves credit assignment and facilitates faster reward propagation during Q-learning. Our method demonstrates state-of-the-art performance across the D4RL suite, particularly excelling in long-horizon, sparse-reward tasks.

## 1 INTRODUCTION

Offline reinforcement learning (RL) offers a promising approach to learning policies from static datasets. These datasets are often comprised of undirected demonstrations and suboptimal sequences collected using different *behavior policies*. Several methods (Fujimoto et al. (2019); Kostrikov et al.; Kumar et al. (2020)) have been proposed for offline RL, all of which aim to strike a balance between constraining the learned policy to the support of the behavior policy and improving upon it. At the core of many of these approaches is an attempt to mitigate the *extrapolation error* which arises while querying the learned Q-function on out-of-support samples for policy improvement. For example, in order to extract the best policy from the data, Q-learning uses an *argmax* over actions to obtain the temporal-difference target. However, querying the Q-function on out-of-support state-actions can lead to errors via exploiting an imperfect Q-function (Fujimoto et al. (2019)).

Framing offline RL as a generative modeling problem has gained significant traction (Chen et al. (2021); Janner et al. (2021)); however, the performance is dependent on the power of the generative models used. These methods either avoid learning a Q-function or rely on other offline Q-learning methods. Recently diffusion models (Sohl-Dickstein et al. (2015); Song & Ermon (2019)), have emerged as state-of-the-art generative models for conditional image-generation (Ramesh et al. (2022); Saharia et al. (2022a)). **Rather than avoiding Q-learning, we model the behavioral policy with diffusion and use this to avoid extrapolation error through batch-constraining.** Previous diffusion-based sequence modelling methods in offline RL diffused over the raw state-action space. However, the low-level trajectory space tends to be poorly suited for reasoning. Some prior works (Pertsch et al. (2021); Ajay et al. (2020)) have proposed to instead reason in more well-conditioned spaces composed of higher-level behavioral primitives. Such temporal abstraction has been shown to result in faster and more reliable credit assignment (Machado et al. (2023); Mann & Mannor (2014)), particularly in long-horizon sparse-reward tasks. **We harness the expressivity of powerful diffusion generative models to reason with temporal abstraction and improve credit assignment.**

Inspired by the recent successes of Latent Diffusion Models (LDMs) (Rombach et al. (2022); Jun & Nichol (2023)), we propose learning similar latent trajectory representations for offline RL tasks by encoding rich high-level behaviors and learning a policy decoder to roll out low-level action sequences conditioned on these behaviors. The idea is to diffuse over semantically rich latent representations while relying on powerful decoders for high-frequency details. Prior works which explored diffusion for offline RL (Janner et al. (2022), Ajay et al.) directly diffused over the raw state-action space, and their architectural considerations for effective diffusion models limited the networks to be simple U-Nets (Ronneberger et al. (2015)). The separation of the diffusion model from the low-level policy allows us to model the low-level policy using a powerful autoregressive decoder. We perform state-conditioned latent diffusion on the learnt latent space and then learn a Q-function over states and corresponding latents. During Q-learning, we batch-constrain the candidate latents for the target Q-function using our expressive diffusion prior, thus mitigating extrapolation error. Our final policy samples latent skills from the LDM, scores the latents using the Q-function and executes the best behavior with the policy decoder. We refer to our method as **L**atent **D**iffusion-**C**onstrained **Q**-learning (LDCQ).

There proposed latent diffusion skill learning method offers several advantages:

**Flexible decoders for high-fidelity actions.** The latent diffusion framework allows us to use powerful decoders for our low-level policy $\pi_\theta$. Previous diffusion works for offline RL (Janner et al. (2022), Ajay et al.) directly diffused over the raw state-action space, and architectural considerations for effective diffusion models limited the networks to be simple U-Nets (Ronneberger et al. (2015)). The separation of the diffusion model from the low-level policy allows us to model $\pi_\theta$ using an expressive autoregressive decoder. (Model architecture discussed in Appendix A.2). We also note that LDMs can be easily used to model trajectories with discrete action spaces, since the decoder and diffusion models are separated.

**Temporal Abstraction with information dense latent space.** Prior works (Pertsch et al. (2021); Ajay et al. (2020)) have learned latent space representations of skills using VAEs. Their use of weaker Gaussian priors forces them to use higher values of the KL penalty multiplier $\beta$ to ensure the latents are well regularized. However, doing so restricts the information capacity of the latent, which limits the variation in behaviors captured by the latents. As we show in section 5.1, increasing the horizon $H$ reveals a clear separation of useful behavioral modes when the latents are weakly constrained. Our method allows modeling the dense latent space with diffusion.

**Faster training and inference.** Inference with LDMs is significantly faster than having to reconstruct the entire trajectory every forward pass with a raw trajectory diffusion model. The training process is also more memory efficient since the networks can be much smaller.

Our method excels at long-horizon credit assignment through temporal abstraction, which allows it to outperform prior offline RL algorithms in the challenging sparse reward antmaze and franka-kitchen tasks. Further, the expressiveness of diffusion models also facilitates greatly improved batch-constrained Q-learning outperforming prior work in the Adroit suite and the image-based Carla lane driving task. Through these strong results, we show that **Batch-Constrained Q-learning is a much stronger method than prior work has indicated, when paired with more expressive generative models and temporal abstraction.** We also demonstrate how LDCQ can be extended to goal-conditioned reinforcement learning.

## 2 RELATED WORK

**Offline RL.** Offline RL poses the challenge of distributional shift while stitching suboptimal trajectories together. Conservative Q-Learning (CQL) (Kumar et al. (2020)) tries to constrain the policy to the behavioral support by learning a pessimistic Q-function that lower-bounds the optimal value function. Implicit Q-Learning (IQL) (Vieillard et al. (2022)) tries to avoid extrapolation error by performing a trade-off between SARSA and DQN using expectile regression. Inspired by notable achievements of generative models in various domains including text-generation (Vaswani et al. (2017)), speech synthesis (Kong et al.) and image-generation (Ramesh et al. (2022); Saharia et al. (2022a)), Chen et al. (2021) proposed to use a generative model for offline RL and bypass the need for Q-learning or bootstrapping altogether with *return-conditioning* (Srivastava et al. (2019); Kumar et al. (2019)). Our method instead formulates a solution with batch-constraining which uses

generative models to model the data distribution and use it to generate candidate actions to learn a Q-function without extrapolation-error. This relies on the assumption that sampling from the generative model does not sample out-of-support samples, which has been difficult to achieve with previously used generative models in offline RL. This is a form of Batch-Constrained Q-Learning (BCQ) (Fujimoto et al. (2019)). Further, to effectively address the problem of stitching, Pertsch et al. (2021) and Ajay et al. (2020) proposed learning policies in latent-trajectory spaces. However, they have to rely on a highly constrained latent space which is not rich enough for the downstream policy. Our proposed method to use latent diffusion, which can model complex distributions, allows for the needed flexibility in the latent space for effective Q-learning and the final policy.

**Diffusion Probabilistic Models.** Diffusion models (Sohl-Dickstein et al. (2015); Song & Ermon (2019)) have emerged as state-of-the-art generative models for conditional image-generation (Ramesh et al. (2022); Saharia et al. (2022a)), super-resolution (Saharia et al. (2022b)) and inpainting (Lugmayr et al. (2022)). Recent offline RL works (Janner et al. (2022), Ajay et al.) have proposed using diffusion to model trajectories and showcased its effectiveness in stitching together behaviors to improve upon suboptimal demonstrations. However, Janner et al. (2022) make the assumption that the value function is learnt using other offline methods and their classifier-guided diffusion requires querying the value function on noisy samples, which can lead to extrapolation-error. Similarly, Ajay et al. can suffer from distributional shift, as it relies on return-conditioning, and maximum returns from arbitrary states can be unknown without access to a value function. We propose a method for learning Q-functions in latent trajectory space with latent diffusion while avoiding extrapolation-error and facilitating long horizon trajectory stitching and credit assignment.

# 3 PRELIMINARIES

## 3.1 DIFFUSION PROBABILISTIC MODELS

Diffusion models (Sohl-Dickstein et al. (2015); Song & Ermon (2019)) are a class of latent variable generative model which learn to generate samples from a probability distribution $p(\mathbf{x})$ by mapping Gaussian noise to the target distribution through an iterative process. They are of the form $p_\psi(\mathbf{x}_0) := \int p_\psi(\mathbf{x}_{0:T}) d\mathbf{x}_{1:T}$ where $\mathbf{x}_0, \ldots \mathbf{x}_T$ are latent variables and the model defines the approximate posterior $q(\mathbf{x}_{1:T} \mid \mathbf{x}_0)$ through a fixed Markov chain which adds Gaussian noise to the data according to a variance schedule $\beta_1, \ldots, \beta_T$. This process is called the *forward diffusion process*:

$$q(\mathbf{x}_{1:T} \mid \mathbf{x}_0) := \prod_{t=1}^{T} q(\mathbf{x}_t \mid \mathbf{x}_{t-1}), \qquad q(\mathbf{x}_t \mid \mathbf{x}_{t-1}) := \mathcal{N}(\mathbf{x}_t; \sqrt{1-\beta_t}\mathbf{x}_{t-1}, \beta_t \mathbf{I}) \quad (1)$$

The forward distribution can be computed for an arbitrary timestep $t$ in closed form. Let $\alpha_t = 1 - \beta_t$ and $\bar{\alpha}_t = \prod_{i=1}^{t} \alpha_i$. Then $q(\mathbf{x}_t \mid \mathbf{x}_0) = \mathcal{N}(\mathbf{x}_t; \sqrt{\bar{\alpha}_t}\mathbf{x}_0, (1-\bar{\alpha}_t)\mathbf{I})$.

Diffusion models learn to sample from the target distribution $p(\mathbf{x})$ by starting from Gaussian noise $p(\mathbf{x}_T) \sim \mathcal{N}(\mathbf{0}, \mathbf{I})$ and iteratively *denoising* the noise to generate in-distribution samples. This is defined as the *reverse diffusion process* $p_\psi(\mathbf{x}_{t-1} \mid \mathbf{x}_t)$:

$$p_\psi(\mathbf{x}_{0:T}) := p(\mathbf{x}_T) \prod_{t=1}^{T} p_\psi(\mathbf{x}_{t-1} \mid \mathbf{x}_t), \qquad p_\psi(\mathbf{x}_{t-1} \mid \mathbf{x}_t) := \mathcal{N}(\mathbf{x}_{t-1}; \mu_\psi(\mathbf{x}_t, t), \mathbf{\Sigma}_\psi(\mathbf{x}_t, t)) \quad (2)$$

The reverse process is trained by minimizing a surrogate loss-function (Ho et al. (2020)):

$$\mathcal{L}(\psi) = \mathbb{E}_{t \sim [1,T], \mathbf{x}_0 \sim q(\mathbf{x}_0), \epsilon \sim \mathcal{N}(\mathbf{0}, \mathbf{I})} \| \epsilon - \epsilon_\psi(\mathbf{x}_t, t) \|^2 \quad (3)$$

Diffusion can be performed in a compressed latent space $\mathbf{z}$ (Rombach et al. (2022)) instead of the final high-dimensional output space of $\mathbf{x}$. This separates the reverse diffusion model $p_\psi(\mathbf{z}_{t-1} \mid \mathbf{z}_t)$ from the decoder $p_\theta(\mathbf{x} \mid \mathbf{z})$. The training is done in two stages, where the decoder is jointly trained with an encoder, similar to a $\beta$-Variational Autoencoder (Kingma & Welling; Higgins et al. (2016)) with a low $\beta$. The prior is then trained to fit the optimized latents of this model. We explain this two-stage training in more detail in section 4.1.

## 3.2 Offline Reinforcement Learning

The reinforcement learning (RL) problem can be formulated as a Markov decision process (MDP). This MDP is a tuple $\langle \rho_0, \mathcal{S}, \mathcal{A}, r, P, \gamma \rangle$, where $\rho_0$ is the initial state distribution, $\mathcal{S}$ is a set of states, $\mathcal{A}$ is a set of actions, $r : \mathcal{S} \times \mathcal{A} \to \mathbb{R}$ is the reward function, $P : \mathcal{S} \times \mathcal{A} \times \mathcal{S} \to [0, 1]$ is the transition function that defines the probability of moving from one state to another after taking an action, and $\gamma \in [0, 1)$ is the discount factor that determines the importance of future rewards. The goal in RL is to learn a policy, i.e., a mapping from states to actions, that maximizes the expected cumulative discounted reward. In the offline RL setting (Levine et al., 2020), the agent has access to a static dataset $\mathcal{D} = \{\mathbf{s}_t^i, \mathbf{a}_t^i, \mathbf{s}_{t+1}^i, r_t^i\}$ of transitions generated by a unknown behavior policy $\pi_\beta(\mathbf{a} \mid \mathbf{s})$ and the goal is to learn a new policy using only this dataset without interacting with the environment. Unlike behavioral cloning, offline RL methods seek to improve upon the behavior policy used to collect the offline dataset. The distribution mismatch between the behavior policy and the training policy can result in problems such as querying the target Q-function with actions not supported in the offline dataset leading to the extrapolation error problem.

## 4 Latent Diffusion Reinforcement Learning

In this section, we elaborate on our latent diffusion-based method for offline RL.

### 4.1 Two-Stage LDM training

**Latent Representation and Low-Level Policy.** The first stage in training the latent diffusion model is comprised of learning a latent trajectory representation. Given a dataset $\mathcal{D}$ of $H$-length trajectories $\boldsymbol{\tau}_H$ represented as sequences of states and actions, $\mathbf{s}_0, \mathbf{a}_0, \mathbf{s}_1, \mathbf{a}_1, \cdots \mathbf{s}_{H-1}, \mathbf{a}_{H-1}$, we want to learn a low-level policy $\pi_\theta(\mathbf{a}|\mathbf{s}, \mathbf{z})$ such that $\mathbf{z}$ represents high-level behaviors in the trajectory. This is done using a $\beta$-Variational Autoencoder (VAE) (Kingma & Welling; Higgins et al. (2016)). Specifically, we maximize the evidence lower bound (ELBO):

$$\mathcal{L}(\theta, \phi, \omega) = \mathbb{E}_{\boldsymbol{\tau}_H \sim D}[\mathbb{E}_{q_\phi(\mathbf{z}|\boldsymbol{\tau}_H)}[\sum_{t=0}^{H-1} \log \pi_\theta(\mathbf{a}_t \mid \mathbf{s}_t, \mathbf{z})] - \beta D_{KL}(q_\phi(\mathbf{z} \mid \boldsymbol{\tau}_H) \mid\mid p_\omega(\mathbf{z} \mid \mathbf{s}_0))] \quad (4)$$

where $q_\phi$ represents our approximate posterior over $\mathbf{z}$ given $\boldsymbol{\tau}_H$, and $p_\omega$ represents our conditional Gaussian prior over $\mathbf{z}$, given $\mathbf{s}_0$. Note that unlike BCQ (Fujimoto et al. (2019)), which uses the VAE as the generative model, we only use the $\beta$-VAE to learn a latent space to diffuse over. As such, the conditional Gaussian prior $p_\omega$ is simply a loose regularization of this latent space, and only weakly constrains the posterior. This is facilitated by the ability of latent diffusion models to later sample from such complex latent distributions. Prior works (Pertsch et al. (2021); Ajay et al. (2020)) have learned latent space representations of skills using VAEs. Their use of weaker Gaussian priors forces them to use higher values of the KL penalty multiplier $\beta$ to ensure the latents are well regularized. However, doing so restricts the information capacity of the latent, which limits the variation in behaviors captured by the latents. As we show in Section 5.1, increasing the horizon $H$ reveals a clear separation of useful behavioral modes when the latents are weakly constrained.

The use of latent diffusion gives us flexibility to make the decoder more powerful. The low-level policy $\pi_\theta$ is represented as an autoregressive model which can capture the fine details across the action dimensions, and is similar to the decoders used by Ghasemipour et al. (2021) and Ajay et al. (2020). While the environments we test in this work use continuous action spaces, the use of latent diffusion allows the method to easily translate to discrete action spaces too, since the decoder can simply be altered to output a categorical distribution while the diffusion process remains unchanged.

**Latent Diffusion Prior.** For training the diffusion model, we collect a dataset of state-latent pairs $(\mathbf{s}_0, \mathbf{z})$ such that $\boldsymbol{\tau}_H \sim \mathcal{D}$ is a $H$-length trajectory snippet, $\mathbf{z} \sim q_\phi(\mathbf{z} \mid \boldsymbol{\tau}_H)$ where $q_\phi$ is the VAE encoder trained earlier, and $\mathbf{s}_0$ is the first state in $\boldsymbol{\tau}_H$. We want to model the prior $p(\mathbf{z} \mid \mathbf{s}_0)$, which is the distribution of the learnt latent space $\mathbf{z}$ conditioned on a state $\mathbf{s}_0$. This effectively represents the different behaviors possible from the state $\mathbf{s}_0$ as supported by the behavioral policy that collected the dataset. To this end, we learn a conditional latent diffusion model $p_\psi(\mathbf{z} \mid \mathbf{s}_0)$ by learning the time-dependent denoising function $\mu_\psi(\mathbf{z}_t, \mathbf{s}_0, t)$, which takes as input the current diffusion latent estimate $\mathbf{z}_t$ and the diffusion timestep $t$ to predict the original latent $\mathbf{z}_0$. Like Ramesh et al. (2022)

and Jun & Nichol (2023), we found predicting the original latent $\mathbf{z}_0$ works better than predicting the noise $\boldsymbol{\epsilon}$. We reweigh the objective based on the noise level according to Min-SNR-$\gamma$ strategy (Hang et al. (2023)). This re-balances the objective, which is dominated by the loss terms corresponding to diffusion time steps closer to $T$. Concretely, we modify the objective in Eq. 3 to minimize:

$$\mathcal{L}(\psi) = \mathbb{E}_{t \sim [1,T], \boldsymbol{\tau}_H \sim \mathcal{D}, \mathbf{z}_0 \sim q_\phi(\mathbf{z}|\boldsymbol{\tau}_H), \mathbf{z}_t \sim q(\mathbf{z}_t|\mathbf{z}_0)} [\min\{\text{SNR}(t), \gamma\}(\|\, \mathbf{z}_0 - \mu_\psi(\mathbf{z}_t, \mathbf{s}_0, t) \,\|^2)] \quad (5)$$

Note that $q_\phi(\mathbf{z} \mid \boldsymbol{\tau}_H)$ is different from $q(\mathbf{z}_t \mid \mathbf{z}_0)$, the former being the approximate posterior of the trained VAE, while the latter is the forward Gaussian diffusion noising process. We use DDPM (Ho et al. (2020)) to sample from the diffusion prior in this work due to its simple implementation. As proposed in Ho & Salimans, we use classifier-free guidance for diffusion. This modifies the original training setup to learn both a conditional $\mu_\psi(\mathbf{z}_t, \mathbf{s}_0, t)$ and an unconditional model. The unconditional version, is represented as $\mu_\psi(\mathbf{z}_t, \emptyset, t)$ where a dummy token $\emptyset$ takes the place of $\mathbf{s}_0$. The following update is then used to generate samples: $\mu_\psi(\mathbf{z}_t, \emptyset, t) + w(\mu_\psi(\mathbf{z}_t, \mathbf{s}_0, t) - \mu_\psi(\mathbf{z}_t, \emptyset, t))$, where $w$ is a tunable hyper-parameter. Increasing $w$ results in reduced sample diversity, in favor of samples with high conditional density. We summarize the two stage LDM training in Appendix 2.

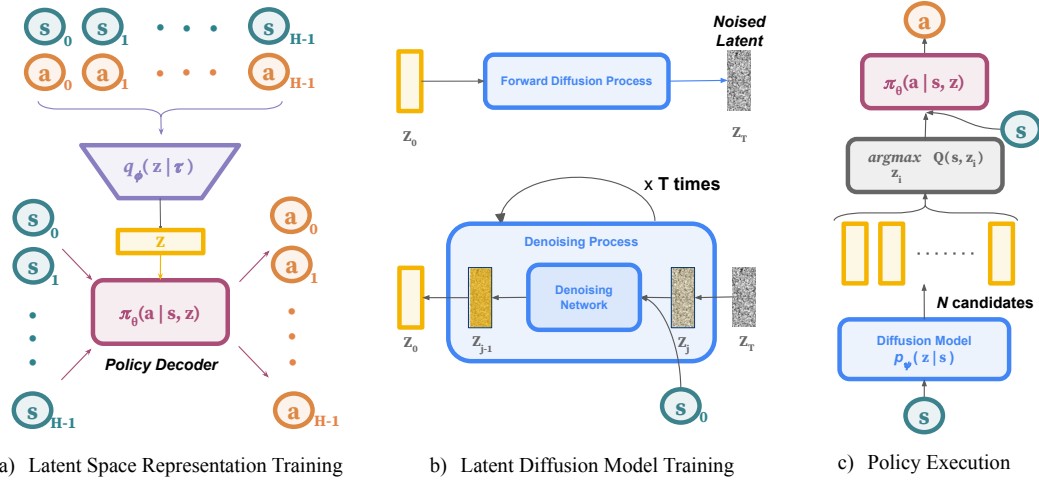

a) Latent Space Representation Training     b) Latent Diffusion Model Training     c) Policy Execution

Figure 1: **Latent Diffusion Reinforcement Learning Overview** a) We first learn the latent space and low-level policy decoder by training a $\beta$-VAE over $H$-length sequences from the demonstrator dataset. b) We train a latent diffusion prior conditioned on $\mathbf{s}_0$ to predict latents generated by the VAE encoder. c) After learning a Q function using LDCQ (Algorithm 1), we score latents sampled by the prior with this Q function and execute the low-level policy $\pi_\theta$ conditioned on the argmax latent.

## 4.2 LATENT DIFFUSION-CONSTRAINED Q-LEARNING (LDCQ)

In batch-constrained Q-learning (BCQ), the target Q-function is constrained to only be maximized using actions that were taken by the demonstrator from the given state (Fujimoto et al. (2019)).

$$\pi(\mathbf{s}) = \underset{\substack{\mathbf{a} \\ s.t.(\mathbf{s},\mathbf{a}) \in \mathcal{D}}}{\text{argmax}}\ Q(\mathbf{s}, \mathbf{a}) \quad (6)$$

In a deterministic MDP setting, BCQ is theoretically guaranteed to converge to the optimal batch-constrained policy. In any non-trivial setting, constraining the policy to actions having support from a given state in the dataset is not feasible, especially if the states are continuous. Instead, a behavior model $\pi_\psi(\mathbf{a} \mid \mathbf{s})$ must be learned on the demonstrator data and samples from this model are used as candidates for the argmax:

$$\pi(\mathbf{s}) = \underset{\mathbf{a}_i \sim \pi_\psi(\mathbf{a}|\mathbf{s})}{\text{argmax}}\ Q(\mathbf{s}, \mathbf{a}_i) \quad (7)$$

However, in many offline RL datasets, the behavior policy is highly multimodal either due to the demonstrations being undirected or because the behavior policy is actually a mixture of unimodal

policies, making it difficult for previously used generative models like VAEs to sample from the distribution accurately. The multimodality of this policy is further exacerbated with increases in temporal abstraction in the latent space, as we show in section 5.1. We propose using latent diffusion to model this distribution, as diffusion is well suited for modelling such multi-modal distributions. We propose to learn a Q-function in the latent action space with latents sampled from the diffusion model. Specifically, we learn a Q-function $Q(\mathbf{s}, \mathbf{z})$, which represents the action-value of a latent action sequence $\mathbf{z}$ given state $\mathbf{s}$. At test time, we generate candidate latents from the diffusion prior $p_\psi(\mathbf{z} \,|\, \mathbf{s})$ and select the one which maximizes the learnt Q-function. We then use this latent with the low-level policy $\pi_\theta(\mathbf{a}_i \,|\, \mathbf{s}_i, \mathbf{z})$ to generate the action sequence for $H$ timesteps.

**Training.** We collect a replay buffer $\mathcal{B}$ for the dataset $\mathcal{D}$ of $H$-length trajectories and store transition tuples $(\mathbf{s}_t, \mathbf{z}, r_{t:t+H}, \mathbf{s}_{t+H})$ from $\boldsymbol{\tau}_H \sim \mathcal{D}$, where $\mathbf{s}_t$ is the first state in $\boldsymbol{\tau}_H$, $\mathbf{z} \sim q_\phi(\mathbf{z} \mid \boldsymbol{\tau}_H)$ is the latent sampled from the VAE approximate posterior, $r_{t:t+H}$ represents the $\gamma$-discounted sum of rewards accumulated over the $H$ time-steps in $\boldsymbol{\tau}_H$, and $\mathbf{s}_{t+H}$ represents the state at the end of $H$-length trajectory snippet. The Q-function is learned with temporal-difference updates (Sutton & Barto (2018)), where we sample a batch of latents for the target argmax using the diffusion prior $p_\psi(\mathbf{z} \,|\, \mathbf{s}_{t+H})$. This should only sample latents which are under the support of the behavioral policy, and hence with the right temporal abstraction, allows for stitching skills to learn an optimal policy grounded on the data support. The resulting Q update can be summarized as:

$$Q(\mathbf{s}_t, \mathbf{z}) \leftarrow (r_{t:t+H} + \gamma^H Q(\mathbf{s}_{t+H}, \operatorname*{argmax}_{\mathbf{z}_i \sim p_\psi(\mathbf{z}|\mathbf{s}_{t+H})} (Q(\mathbf{s}_{t+H}, \mathbf{z}_i)))) \tag{8}$$

We use Clipped Double Q-learning as proposed in (Fujimoto et al. (2018)) to further reduce over-estimation bias during training. We also use Prioritized Experience Replay (Schaul et al. (2015)) to accelerate the training in sparse-reward tasks like AntMaze and FrankaKitchen. We summarize our proposed LDCQ method in Algorithm 1.

---

**Algorithm 1** Latent Diffusion-Constrained Q-Learning (LDCQ)

---

1: **Input:** prioritized-replay-buffer $\mathcal{B}$, horizon $H$, target network update-rate $\rho$, mini-batch size $N$, number of sampled latents $n$, maximum iterations $M$, discount-factor $\gamma$, latent diffusion denoising function $\mu_\psi$, variance schedule $\alpha_1, \ldots, \alpha_T, \bar{\alpha}_1, \ldots, \bar{\alpha}_T, \beta_1, \ldots, \beta_T$.
2: Initialize Q-networks $Q_{\Theta_1}$ and $Q_{\Theta_2}$ with random parameters $Q_{\Theta_1}, Q_{\Theta_2}$ and target Q-networks $Q_{\Theta_1^{target}}$ and $Q_{\Theta_2^{target}}$ with $\Theta_1^{target} \leftarrow \Theta_1, \Theta_2^{target} \leftarrow \Theta_2$
3: **for** $iter = 1$ to $M$ **do**
4:      Sample a minibatch of $N$ transitions $\{(\mathbf{s}_t, \mathbf{z}, r_{t:t+H}, \mathbf{s}_{t+H})\}$ from $\mathcal{B}$
5:      Sample $n$ latents for each transition: $\mathbf{z}_T \sim \mathcal{N}(\mathbf{0}, \mathbf{I})$
6:      **for** $t = T$ to $1$ **do**                                        ▷ DDPM Sampling
7:          $\hat{\mathbf{z}} = \mu_\psi(\mathbf{z}_t, \varnothing, t) + w(\mu_\psi(\mathbf{z}_t, \mathbf{s}_{t+H}, t) - \mu_\psi(\mathbf{z}_t, \varnothing, t))$
8:          $\mathbf{z}_{t-1} \sim \mathcal{N}(\frac{\sqrt{\alpha_t}(1-\bar{\alpha}_{t-1})}{1-\bar{\alpha}_t}\mathbf{z}_t + \frac{\sqrt{\bar{\alpha}_{t-1}}\beta_t}{1-\bar{\alpha}_t}\hat{\mathbf{z}}, \mathbb{I}(t > 1)\beta_t\mathbf{I})$
9:      **end for**
10:      Compute the target values $y = r_{t:t+H} + \gamma^H \{\max_{\mathbf{z}_0}\{\min_{j=1,2} Q_{\Theta_j^{target}}(\mathbf{s}_{t+H}, \mathbf{z}_0)\}\}$
11:      Update $Q$-networks by minimizing the loss: $\frac{1}{N}||y - Q_\Theta(\mathbf{s}_t, \mathbf{z})||_2^2$
12:      Update target $Q$-networks: $\Theta^{target} \leftarrow \rho\Theta + (1 - \rho)\Theta^{target}$
13: **end for**

---

**Policy Execution.** The final policy for LDCQ comprises generating candidate latents $\mathbf{z}$ for a particular state $\mathbf{s}$ using the latent diffusion prior $\mathbf{z} \sim p_\psi(\mathbf{z} \mid \mathbf{s})$. These latents are then scored using the learnt Q-function and the best latent $\mathbf{z}_{max}$ is decoded using the VAE autoregressive decoder $\mathbf{a} \sim \pi_\theta(\mathbf{a} \mid \mathbf{s}, \mathbf{z}_{max})$ to obtain H-length action sequences which are executed sequentially. Note that the latent diffusion model is used both during training the Q-function and during the final evaluation phase, ensuring that the sampled latents do not go out-of-support. The policy execution algorithm is detailed in the Appendix 3.

### 4.3 LATENT DIFFUSION GOAL CONDITIONING (LDGC)

Diffuser (Janner et al. (2022)) proposed framing certain navigation problems as a sequence inpainting task, where the last state of the diffused trajectory is set to be the goal during sampling. We

can similarly condition our diffusion prior on the goal to sample from feasible latents that lead to the goal. This prior is of the form $p_\psi(\mathbf{z} \mid \mathbf{s}_0, \mathbf{s}_g)$, where $\mathbf{s}_g$ is the target goal state. Since with latent diffusion, the training of the low-level policy alongside the VAE is done separately from the diffusion prior training, we can reuse the same VAE posterior to train different diffusion models, such as this goal-conditioned variant. At test time, we perform classifer-free guidance to further push the sampling towards high-density goal-conditioned latents. For tasks which are suited to goal conditioning, this can be simpler to implement and achieves better performance than Q-learning. Also, unlike Diffuser, our method does not need to have the goal within the planning horizon of the trajectory. This allows our method to be used for arbitrarily long-horizon tasks.

## 5 EXPERIMENTAL EVALUATION AND ANALYSIS

In our experiments, we focus on **1)** studying the helpfulness temporal abstraction has in distinguishing latent skills (Section 5.1) **2)** evaluating the ability of diffusion models to sample from the latent space (section 5.2 and 5.3) and **3)** evaluating the performance of our method in the D4RL offline RL benchmarks (section 5.4).

### 5.1 TEMPORAL ABSTRACTION INDUCES MULTI-MODALITY IN LATENT SPACE

In this section, we study how the horizon length $H$ affects the latent space and provide empirical justification for learning long-horizon latent space representations. For our experiment, we consider the *kitchen-mixed-v0* task from the D4RL benchmark suite (Fu et al. (2020)). The goal in this task is to control a 9-DoF robotic arm to manipulate multiple objects (microwave, kettle, burner and a switch) sequentially, in a single episode to reach a desired configuration, with only sparse 0-1 completion reward for every object that attains the target configuration. As Fu et al. (2020) states, there is a high degree of multi-modality in this task arising from the demonstration trajectories because different trajectories in the dataset complete the tasks in a random order. Thus, before starting to solve any task, the policy implicitly needs to *choose* which task to solve and then generate the actions to solve the task. Given a state, the dataset can consist of multiple behavior modes, and averaging over these modes leads to suboptimal action sequences. Hence, being able to differentiate between these tasks is desirable.

We hypothesize that as we increase our sequence horizon $H$, we should see better separation between the modes. In Figure 2, we plot a 2D (PCA) projection of the VAE encoder latents of the starting state-action sequences in the kitchen-mixed dataset. With a lower horizon, these modes are difficult to isolate and the latents appear to be drawn from a Normal distribution (Figure 2). However, as we increase temporal abstraction from $H = 1$ to $H = 20$, we can see *three* distinct modes emerge, which when cross-referenced with the dataset correspond to the three common tasks executed from the starting state by the behavioral policy (microwave, kettle, and burner). These modes better capture the underlying variation in an action sequence, and having picked one we can run our low-level policy to execute it. As demonstrated in our experiments, such temporal abstraction facilitates easier Q-stitching, with better asymptotic performance. However, in order to train these abstract Q functions, it is necessary to sample from the complex multi-modal distribution and the VAE conditional Gaussian prior is no longer adequate for this purpose, as shown in section 5.2.

### 5.2 LDMS ADDRESS MULTI-MODALITY IN LATENT SPACE

In this section, we provide empirical evidence that latent diffusion models are superior in modelling multi-modal distributions as compared to VAEs. For our experiment, we again consider the *kitchen-mixed-v0* task. The goal of the generative model here is to learn the prior distribution $p(\mathbf{z} \mid \mathbf{s})$ and sample from it such that we can get candidate latents corresponding to state $\mathbf{s}$ belonging to the support of the dataset. However, as demonstrated earlier, the multi-modality in the latent spaces increases with the horizon. We visualize the latents from the initial states of all trajectories in the dataset in Figure 3a using PCA with $H = 20$. The three clusters in the figure correspond to the latents of three different tasks namely microwave, kettle and burner. Similarly, we also visualize the latents predicted by the diffusion model Figure 3b) and the VAE conditional prior Figure 3c) for the same initial states by projecting them onto the principal components of the ground truth latents. We can see that the diffusion prior is able to sample effectively all modes from the ground truth

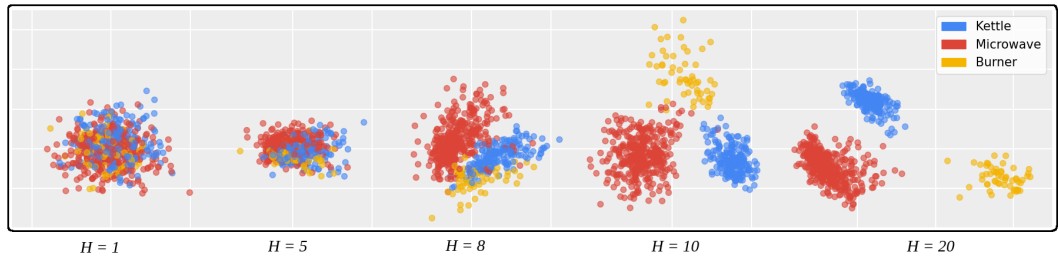

Figure 2: **Projection of latents across horizon**. Latent projections of trajectory snippets with different horizon lengths $H$. From the initial state there are 3 tasks (Kettle, Microwave, Burner) which are randomly selected at the start of each episode. These 3 primary modes emerge as we increase $H$, with the distribution turning multi-modal.

latent distribution, while the VAE prior spreads its mass over the three modes, and thus samples out of distribution in between the three modes. Using latents sampled from the VAE prior to learn the Q-function can thus lead to sampling from out of the support, resulting in extrapolation error.

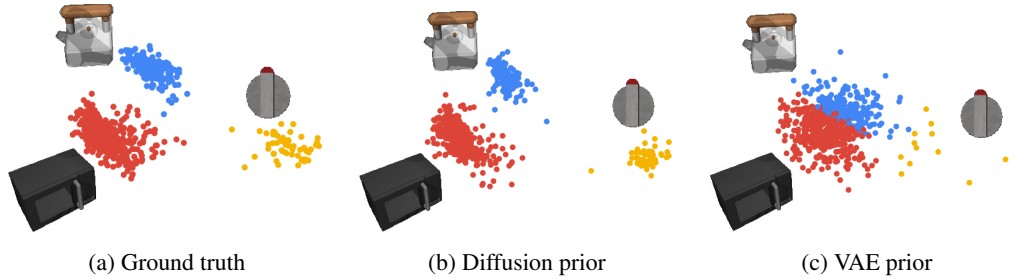

(a) Ground truth      (b) Diffusion prior      (c) VAE prior

Figure 3: Visualization of latents projected using PCA for kitchen-mixed with $H = 20$. The diffusion prior models the ground truth much more accurately while the VAE prior generates out-of-distribution samples.

## 5.3 PERFORMANCE IMPROVEMENT WITH TEMPORAL ABSTRACTION

We empirically demonstrate the importance of temporal abstraction and the performance improvement with diffusion on modelling temporally abstract latent spaces. We compare our method with a variant of BCQ which uses temporal abstraction ($H > 1$), which we refer to as BCQ-H. We use the same VAE architecture here as LDCQ, and fit the conditional Gaussian prior with a network having comparable parameters to our diffusion model. We find that generally, increasing the horizon $H$ results in better performance, both in BCQ-H and LDCQ, and both of them eventually saturate and degrade, possibly due to the limited decoder capacity. With $H = 1$, the latent distribution is roughly Normal as discussed earlier and our diffusion prior is essentially equivalent to the Gaussian prior in BCQ, so we see similar performance. As we increase $H$, however, the diffusion prior is able to efficiently sample from the more complex latent distribution that emerges, which allows the resulting policies to benefit from temporal abstraction. BCQ-H, while also seeing a performance boost with increased temporal abstraction, lags behind LDCQ. We plot D4RL score-vs-$H$ for BCQ-H and LDCQ evaluated on the *kitchen-mixed-v0* task in Figure 4. The benefit of temporal abstraction is ablated for different tasks in Appendix J.

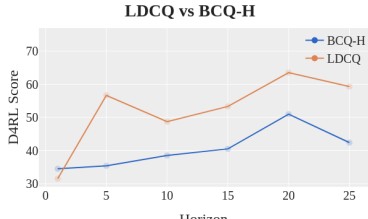

Figure 4: D4RL score of LDCQ and BCQ-H on kitchen-mixed-v0 with varying sequence horizon $H$

## 5.4 OFFLINE RL BENCHMARKS

In this section, we investigate the effectiveness of our Latent Diffusion Reinforcement Learning methods on the D4RL offline RL benchmark suite (Fu et al. (2020)). We compare with Behavior Cloning and several *state-of-the-art* offline RL methods. Diffuser (Janner et al. (2022)) and Decision Diffuser (Ajay et al.) are prior raw trajectory diffusion methods. We found that our method does not require much hyperparameter tuning and only had to vary the sequence horizon $H$ across tasks. In maze2d and AntMaze tasks we use $H = 30$, in kitchen tasks we use $H = 20$ and in locomotion and adroit tasks we use $H = 10$. We train our diffusion prior with $T = 200$ diffusion steps. The other hyperparameters which are constant across tasks are provided in the supplemental material. In Table 1, we compare performance across tasks in the D4RL suite. We would like to highlight tasks in Maze2d, AntMaze and FrankaKitchen environments which are known to be the most difficult in D4RL, with most algorithms performing poorly. Maze2d and AntMaze consist of undirected demonstrations controlling the agent to navigate to random locations in a maze. AntMaze is quite difficult because the agent must learn the high-level trajectory stitching task alongside low-level control of the ant robot with 8-DoF. In the maze navigation tasks, we also evaluate the performance of our goal-conditioned (LDGC) variant. For Diffuser runs we use the goal-conditioned inpainting version proposed by the authors since the classifier-guided version yielded poor results.

Table 1: Performance comparison on D4RL tasks. LDGC evaluated in goal-directed maze tasks.

| Dataset | BC | BCQ | CQL | IQL | DT | Diffuser | DD | LDCQ (Ours) | LDGC (Ours) |
|---|---|---|---|---|---|---|---|---|---|
| maze2d-umaze-v1 | 3.8 | 12.8 | 5.7 | 47.4 | 27.3 | 113.5 | - | **134.2** ± 4.0 | **141.0** ± 2.7 |
| maze2d-medium-v1 | 30.3 | 8.3 | 5.0 | 34.9 | 32.1 | 121.5 | - | **125.3** ± 2.5 | **139.9** ± 4.2 |
| maze2d-large-v1 | 5.0 | 6.2 | 12.5 | 58.6 | 18.1 | 123.0 | - | **150.1** ± 2.9 | **206.8** ± 3.1 |
| antmaze-umaze-diverse-v2 | 45.6 | 55.0 | **84.0** | 62.2 | 54.0 | - | - | 81.4 ± 3.5 | **85.6** ± 2.4 |
| antmaze-medium-diverse-v2 | 0.0 | 0.0 | 53.7 | **70.0** | 0.0 | 45.5 | 24.6 | 68.9 ± 0.7 | **75.6** ± 0.9 |
| antmaze-large-diverse-v2 | 0.0 | 2.2 | 14.9 | 47.5 | 0.0 | 22.0 | 7.5 | 57.7 ± 1.8 | **73.6** ± 1.3 |
| kitchen-complete-v0 | **65.0** | 52.0 | 43.8 | **62.5** | - | - | - | 62.5 ± 2.1 | - |
| kitchen-partial-v0 | 38.0 | 31.7 | 50.1 | 46.3 | 42.0 | - | 57.0 | **67.8** ± 0.8 | - |
| kitchen-mixed-v0 | 51.5 | 34.5 | 52.4 | 51.0 | 50.7 | - | **65.0** | 62.3 ± 0.5 | - |
| halfcheetah-medium-expert-v2 | 55.2 | 64.7 | **91.6** | 86.7 | 86.8 | 88.9 | 90.6 | 90.2 ± 0.9 | - |
| walker2d-medium-expert-v2 | 107.5 | 57.5 | **108.8** | 109.6 | 108.1 | 106.9 | 108.8 | 109.3 ± 0.4 | - |
| hopper-medium-expert-v2 | 52.5 | **110.9** | 105.4 | 91.5 | 107.6 | 103.3 | 111.8 | 111.3 ± 0.2 | - |
| halfcheetah-medium-v2 | 42.6 | 40.7 | 44.0 | 47.4 | 42.6 | 42.8 | **49.1** | 42.8 ± 0.7 | - |
| walker2d-medium-v2 | 75.3 | 53.1 | 72.5 | 78.3 | 74.0 | 79.6 | **82.5** | 69.4 ± 3.5 | - |
| hopper-medium-v2 | 52.9 | 54.5 | 58.5 | 66.3 | 67.6 | 74.3 | **79.3** | 66.2 ± 1.7 | - |
| halfcheetah-medium-replay-v2 | 36.6 | 38.2 | **45.5** | **44.2** | 36.6 | 37.7 | 39.3 | 41.8 ± 0.4 | - |
| walker2d-medium-replay-v2 | 26.0 | 15.0 | **77.2** | 73.9 | 66.6 | 70.6 | 75.0 | 68.5 ± 4.3 | - |
| hopper-medium-replay-v2 | 18.1 | 33.1 | 95.0 | 94.7 | 82.7 | 93.6 | **100.0** | 86.2 ± 2.5 | - |
| pen-human | 34.4 | 68.9 | 37.5 | 71.5 | - | - | - | **74.1** ± 2.7 | - |
| hammer-human | 1.2 | 0.3 | **4.4** | 1.4 | - | - | - | 1.5 ± 0.8 | - |
| door-human | 0.5 | 0.0 | 9.9 | 4.3 | - | - | - | **11.8** ± 1.9 | - |
| relocate-human | 0.0 | -0.1 | 0.2 | 0.1 | - | - | - | **0.3** ± 0.1 | - |
| pen-cloned | 37.0 | 44.0 | 39.2 | 37.3 | - | - | - | **47.7** ± 1.9 | - |
| hammer-cloned | 0.6 | 0.4 | 2.1 | 2.1 | - | - | - | **2.8** ± 1.2 | - |
| door-cloned | 0.0 | 0.0 | 0.4 | **1.6** | - | - | - | 1.1 ± 0.4 | - |
| relocate-cloned | -0.3 | -0.3 | -0.1 | -0.2 | - | - | - | -0.2 ± 0.1 | - |
| pen-expert | 85.1 | 114.9 | 107.0 | - | - | - | - | **121.2** ± 3.6 | - |
| hammer-expert | **125.6** | 107.2 | 86.7 | - | - | - | - | 45.8 ± 10.5 | - |
| door-expert | 34.9 | 99.0 | 101.5 | - | - | - | - | **105.1** ± 0.3 | - |
| relocate-expert | 101.3 | 41.6 | 95.0 | - | - | - | - | **104.7** ± 1.4 | - |
| carla-lane-v0 | 17.2 | -0.1 | 20.9 | 18.6 | - | - | - | **24.7** ± 3.2 | - |

Both LDCQ and LDGC achieve state-of-the-art results in all sparse reward D4RL tasks. The goal-conditioned variant outperforms all others in maze2d and AntMaze. This variant is extremely simple to implement through supervised learning of the diffusion prior with no Q-learning or online planning and is ideal for goal-reaching tasks. We also provide an evaluation of our method on the D4RL locomotion suite and the Adroit robotics suite. While these tasks are not specifically focused on trajectory-stitching, our method is competitive in the locomotion tasks and stronger than baselines in Adroit. To extend our method to Carla's image input spaces, we compress the image using a $\beta$-VAE encoder such that our method operates on a compressed state space (more in Appendix G).

## 6 CONCLUSION

In this work, we showed that offline RL datasets comprised of suboptimal demonstrations have expressive multi-modal latent spaces which can be captured with temporal abstraction and is well suited for learning high-reward policies. With a powerful conditional generative model to capture the richness of this latent space, we demonstrated that the simple batch-constrained Q-learning framework can be directly used to obtain strong performance. Our biggest improvements come from long-horizon sparse reward tasks, which most prior offline RL methods struggled with, even previous raw trajectory diffusion methods. Our approach also required no task-specific tuning, except for the sequence horizon $H$. We believe that latent diffusion has enormous potential in offline RL and our work has barely scratched the surface of possibilities.

## 7 REPRODUCIBILITY

We provide link to our code in section A.1. We provide details of our model architectures in section A.2 and hyperparameters in A.3. Our experiments were conducted on the open D4RL benchmark datasets.

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

# A  Training Details

## A.1  Source Code

The source code is available at: https://github.com/ldcq/ldcq.

## A.2  Network Architecture

### A.2.1  Variational Autoencoder

**Encoder.** For learning the latent trajectory representation, our VAE uses an architecture similar to Ajay et al. (2020). The encoder consists of two stacked bidirectional GRU layers, followed by mean and standard deviation heads which are each a 2 layer MLP with RELU activation for the hidden layers. The mean output head is a linear layer. The standard deviation output head is followed by a SoftPlus activation function to ensure it is always positive. The hidden layer dimension is fixed to 256.

**Decoder.** For the low-level policy decoder, we use an auto-regressive policy network similar to that described in EMAQ (Ghasemipour et al. (2021)), in which each element of the action vector has its own MLP network, taking as input the current state, latent representation, and all previously-sampled action elements. The complete action vector is sampled element-by-element, with the most recently sampled element becoming an input to the network for the next element. These MLP networks consists of 2 layers followed by 2 layers of mean and standard deviation heads similar to the encoder network. The mean output head is a linear layer and the standard deviation output head is followed by a SoftPlus activation. Again, ReLU activation is used after all hidden layer and the hidden dimension is fixed to 256.

### A.2.2  Diffusion Prior

The diffusion prior is a deep ResNet (He et al. (2016)) architecture consisting of 8 residual blocks. It takes as input a vector representing a latent trajectory $\mathbf{z}$ and outputs a denoised version of the latent. The hidden blocks are of dimensions: [128, 32, 16, 8, 16, 32, 128]. Similar to a U-Net (Ronneberger et al. (2015)), the initial blocks are connected by residual connections to the later blocks having the same hidden dimension. The diffusion timestep $t$ is encoded with a 256-dimensional sinusoidal embedding and then further encoded with a 2-layer MLP. The conditioning state $\mathbf{s}$ is also encoded by a 2 layer MLP. In each residual block, the state and time encodings are concatenated with the current layer activation for conditioning. When training the unconditional diffusion model for classifier-free guidance, the state input is given as a vector of zeros to represent a null vector.

### A.2.3  Q-networks

The Q-networks take as input the state $\mathbf{s}$, latent $\mathbf{z}$ and consist of a 5 layer MLP with 256 hidden units in the first 3 layers, 32 hidden units in the third layer, and finally a linear output layer. We use GELU activation function between hidden layers. LayerNorm is applied before each activation.

## A.3  Hyperparameters

The hyperparameters which are constant across tasks for the different stages of our proposed method are given in Tables 2, 3 and 4.

## A.4  Hardware

The models were trained on NVIDIA RTX A6000. Since different tasks have different dataset sizes, the model training times changes across tasks. Depending on the task, training the $\beta$-VAE took between 3-7 hours, the diffusion prior took between 4-12 hours and the Q-Learning took between 3-5 hours.

Table 2: $\beta$-VAE hyperparameters

| Parameter | Value |
|---|---|
| Learning rate | 5e-5 |
| Batch size | 128 |
| Epochs | 100 |
| Latent dimension ($\mathbf{z}$) | 16 |
| $\beta$ | 0.05 |
| Hidden layer dimension | 256 |

Table 3: Diffusion training hyperparameters

| Parameter | Value |
|---|---|
| Learning rate | 1e-4 |
| Batch size | 32 |
| Epochs | 300 |
| Diffusion steps ($T$) | 500 |
| Drop probability (For unconditional prior) | 0.1 |
| Variance schedule | linear |
| Sampling algorithm | DDPM |
| $\gamma$ (For Min-SNR-$\gamma$ weighing) | 5 |

Table 4: Q-Learning hyperparameters

| Parameter | Value |
|---|---|
| Learning rate | 5e-4 |
| Batch size | 128 |
| Discount factor ($\gamma$) | 0.995 |
| Target net update rate ($\rho$) | 0.995 |
| PER buffer $\alpha$ | 0.7 |
| PER buffer $\beta$ | Linearly increased from 0.3 to 1, Grows by 0.03 every 3000 steps |
| Diffusion samples for batch argmax | 500 |

## B  Low Level Policy and Skill Prior Training

We describe the training of the $\beta$-VAE for learning the skill representations and the low level policy, and then training the diffusion skill prior in Algorithm 2

---

**Algorithm 2** LL Policy and Skill Prior Training

---

1: **Input:** horizon $H$, KL regularization coefficient $\beta$, number of steps $M$, diffusion timesteps $T$, $\gamma$ for Min-SNR-$\gamma$, sequence encoder $q_\phi$, conditional Gaussian prior $p_\omega$, policy decoder $\pi_\theta$, latent diffusion prior $\mu_\psi$, offline dataset $\mathcal{D}$, variance schedule $\alpha_1, \ldots, \alpha_T, \bar{\alpha}_1, \ldots, \bar{\alpha}_T, \beta_1, \ldots, \beta_T$.
2: **for** $iter = 1$ to $M$ **do**                                                 ▷ Training $\beta$-VAE
3:      Sample a $n$ size minibatch of $H$ length subtrajectories $\{\mathbf{s}_{t:t+H}^{(1:n)}, \mathbf{a}_{t:t+H}^{(1:n)}\}$ from $\mathcal{D}$
4:      Sample $\mathbf{z}^i \sim q_\phi(\mathbf{z}|\mathbf{s}_{t:t+H}^{(i)})$
5:      $\mathcal{L}(\phi, \theta, \omega) = \sum_{i=1}^{n} \sum_{j=t}^{t+H} -\log \pi_\theta(\mathbf{a}_j^{(i)}|\mathbf{s}_j^{(i)}, \mathbf{z}^{(i)}) + \beta \mathcal{D}_{KL}(q_\phi(\mathbf{z}|\mathbf{s}_{t:t+H}^{(i)})||p_\omega(\mathbf{z}|\mathbf{s}_t^{(i)}))$
6:      Take gradient step to minimize $\mathcal{L}$
7: **end for**
8: **for** $iter = 1$ to $M$ **do**                                         ▷ Training diffusion skill prior
9:      Sample a $n$ size minibatch of $H$ length subtrajectories $\{\mathbf{s}_{t:t+H}^{(1:n)}, \mathbf{a}_{t:t+H}^{(1:n)}\}$ from $\mathcal{D}$
10:     Sample $\mathbf{z}^{(i)} \sim q_\phi(\mathbf{z}|\mathbf{s}_{t:t+H}^{(i)})$
11:     Sample diffusion time $\tau^{(i)} \sim \mathcal{U}[1, T]$
12:     Noise latents $\mathbf{z}_\tau^{(i)} \sim \mathcal{N}(\sqrt{\bar{\alpha}_\tau}\mathbf{z}^{(i)}, (1 - \bar{\alpha}_\tau)\mathbf{I})$
13:     $\mathcal{L}(\psi) = \sum_{i=1}^{n} \min\{\text{SNR}(\tau^{(i)}), \gamma\}(\| \mathbf{z}^{(i)} - \mu_\psi(\mathbf{z}_\tau^{(i)}, \mathbf{s}_t, \tau) \|^2)$
14:     Take gradient step to minimize $\mathcal{L}$
15: **end for**

---

## C  Policy Execution

After training the diffusion prior and Q-learning, we execute the policy as described in Algorithm 3.

---

**Algorithm 3** Policy Execution

---

1: **Input:** horizon $H$, number of latents to sample $n$, Q-function $Q_\Theta$, policy decoder $\pi_\theta$, latent diffusion denoising function $\mu_\psi$, variance schedule $\alpha_1, \ldots, \alpha_T, \bar{\alpha}_1, \ldots, \bar{\alpha}_T, \beta_1, \ldots, \beta_T$.
2: $done = False$
3: **while** not $done$ **do**
4:     Observe environment state $\mathbf{s}_0$
5:     Sample $n$ latents: $\mathbf{z}_T \sim \mathcal{N}(\mathbf{0}, \mathbf{I})$
6:     **for** $t = T$ to 1 **do**                                               ▷ DDPM Sampling
7:        $\hat{\mathbf{z}} = \mu_\psi(\mathbf{z}_t, \varnothing, t) + w(\mu_\psi(\mathbf{z}_t, \mathbf{s}_0, t) - \mu_\psi(\mathbf{z}_t, \varnothing, t))$
8:        $\mathbf{z}_{t-1} \sim \mathcal{N}(\frac{\sqrt{\alpha_t}(1-\bar{\alpha}_{t-1})}{1-\bar{\alpha}_t}\mathbf{z}_t + \frac{\sqrt{\bar{\alpha}_{t-1}}\beta_t}{1-\bar{\alpha}_t}\hat{\mathbf{z}}, \mathbb{I}(t > 1)\beta_t\mathbf{I})$
9:     **end for**
10:    Find best skill by scoring them with Q-function: $i = \underset{i}{\arg\max} \ Q_\Theta(\mathbf{s}_0, \mathbf{z}_0^i)$
11:    $h = 0$
12:    **while** $h < H$ **and** not $done$ **do**                                 ▷ Execute Skill $\mathbf{z}_0^i$
13:        Observe environment state $\mathbf{s}_h$
14:        Get action $\mathbf{a}_h \in \pi_\theta(\mathbf{a}|\mathbf{s}_h, \mathbf{z}_0^i)$
15:        Execute action $\mathbf{a}_h$
16:        Update $done$
17:        $h = h + 1$
18:    **end while**
19: **end while**

---

# D  COMPARISON WITH OTHER LATENT SKILL LEARNING METHODS

In this section, we compare our method against existing algorithms which do latent skill learning with generative models in the D4RL tasks. Specifically, we compare against the VAE based algorithms OPAL (Ajay et al. (2020)) and PLAS (Zhou et al. (2020)), and the Normalizing Flow based algorithm Flow to Control (Yang et al. (2023)). We were not able to replicate the OPAL results with the code provided to us by the authors. We list the best scores we were able to obtain with it. We did not find code to implement Flow to Control, and so use the results listed in their paper. The results are shown in table 5.

Table 5: Performance comparison on D4RL tasks. Algorithms are evaluated only on the tasks listed in their respective papers.

| Dataset | OPAL | PLAS | Flow2Control | LDCQ (Ours) | LDGC (Ours) |
|---|---|---|---|---|---|
| maze2d-large-v1 | - | 122.7 | - | $150.1 \pm 2.9$ | $206.8 \pm 3.1$ |
| antmaze-medium-diverse-v2 | 57.5 | 0.0 | 83.7 | $68.9 \pm 0.7$ | $75.6 \pm 0.9$ |
| antmaze-large-diverse-v2 | 52.0 | 0.0 | 52.8 | $57.7 \pm 1.8$ | $73.6 \pm 1.3$ |
| kitchen-partial-v0 | 55.5 | 43.9 | 74.9 | $67.8 \pm 0.8$ | - |
| kitchen-mixed-v0 | 50.2 | 40.8 | 69.2 | $62.3 \pm 0.5$ | - |
| halfcheetah-medium-expert-v2 | - | 99.3 | - | $90.2 \pm 0.9$ | - |
| walker2d-medium-expert-v2 | - | 97.2 | - | $109.3 \pm 0.4$ | - |
| hopper-medium-expert-v2 | - | 111.0 | - | $111.3 \pm 0.2$ | - |
| halfcheetah-medium-v2 | - | 42.2 | - | $42.8 \pm 0.7$ | - |
| walker2d-medium-v2 | - | 66.9 | - | $69.4 \pm 3.5$ | - |
| hopper-medium-v2 | - | 36.9 | - | $66.2 \pm 1.7$ | - |
| halfcheetah-medium-replay-v2 | - | 45.7 | - | $41.8 \pm 0.4$ | - |
| walker2d-medium-replay-v2 | - | 14.3 | - | $68.5 \pm 4.3$ | - |
| hopper-medium-replay-v2 | - | 51.9 | - | $86.2 \pm 2.5$ | - |
| pen-human | - | 67.3 | 63.1 | $74.1 \pm 2.7$ | - |
| hammer-human | - | 4.6 | 3.3 | $1.5 \pm 0.8$ | - |
| door-human | - | 4.4 | 15.1 | $11.8 \pm 1.9$ | - |
| relocate-human | - | 0.3 | - | $0.3 \pm 0.1$ | - |
| pen-cloned | - | 49.0 | 65.8 | $47.7 \pm 1.9$ | - |
| hammer-cloned | - | 1.0 | 2.1 | $2.8 \pm 1.2$ | - |
| door-cloned | - | 3.3 | 8.1 | $1.1 \pm 0.4$ | - |
| relocate-cloned | - | -0.2 | - | $-0.2 \pm 0.1$ | - |

# E  LATENT DIFFUSION-CONSTRAINED PLANNING (LDCP)

In this section, we explore another method to derive a policy for offline RL with latent diffusion other than our proposed methods *Latent Diffusion-Constrained Q-Learning (LDCQ)* and *Latent Diffusion Goal Conditioning (LDGC)*. This is a model-based method which learns a temporally abstract world model of the environment from offline data. Specifically, we learn a temporally abstract world model $p_\eta(\mathbf{s}_{t+H} \mid \mathbf{s}_t, \mathbf{z})$ that predicts the state outcome of executing a particular latent behavior after $H$ steps. That is, given the current state $\mathbf{s}_t$ and a latent behavior $\mathbf{z}$ the model predicts the distribution of the state $\mathbf{s}_{t+H}$. This is trained in a supervised manner by sampling transition tuples $(\mathbf{s}_t, \mathbf{z}, \mathbf{s}_{t+H})$ from $\boldsymbol{\tau}_H \sim \mathcal{D}$ and minimizing the objective:

$$\mathcal{L}(\eta) = \mathbb{E}_{\boldsymbol{\tau}_H \sim \mathcal{D}} \| p_\eta(\mathbf{s}_{t+H} \mid \mathbf{s}_t, \mathbf{z}) - \mathbf{s}_{t+H} \|^2 \qquad (9)$$

where $\eta$ are the parameters of the temporally abstract world model $p_\eta$.

In goal-reaching environments, we leverage this model to do planning using the diffusion prior. We sample $n$ latents $\mathbf{z}^i$ ($1 \leq i \leq n$) using the diffusion prior for the current state $\mathbf{s}_t$, and use the learnt dynamics model to compute predicted future state $\mathbf{s}_{t+H}^i$ for each latent $\mathbf{z}^i$. These final states are then scored using a cost-function $\mathcal{J}$ and the latent corresponding to the best final state is chosen for

execution. Note that sampling latents from the diffusion prior ensures that the world model is not queried on out-of-support data. We refer to this method as *Latent Diffusion-Constrained Planning (LDCP)*. The planning procedure is summarized in Algorithm 4.

---

**Algorithm 4** Latent Diffusion-Constrained Planning (LDCP)

---

1: **Input:** horizon $H$, number of latents to sample $n$, cost-function $\mathcal{J}$, policy decoder $\pi_\theta$, temporally abstract world model $p_\eta$, latent diffusion denoising function $\mu_\psi$, variance schedule $\alpha_1, \ldots, \alpha_T, \bar{\alpha}_1, \ldots, \bar{\alpha}_T, \beta_1, \ldots, \beta_T$.
2: *done = False*
3: **while** not *done* **do**
4:     Observe environment state $\mathbf{s}_0$
5:     Sample $n$ latents: $\mathbf{z}_T \sim \mathcal{N}(\mathbf{0}, \mathbf{I})$
6:     **for** $t = T$ to 1 **do**                       ▷ DDPM Sampling
7:         $\hat{\mathbf{z}} = \mu_\psi(\mathbf{z}_t, \varnothing, t) + w(\mu_\psi(\mathbf{z}_t, \mathbf{s}_0, t) - \mu_\psi(\mathbf{z}_t, \varnothing, t))$
8:         $\mathbf{z}_{t-1} \sim \mathcal{N}(\frac{\sqrt{\alpha_t}(1-\bar{\alpha}_{t-1})}{1-\bar{\alpha}_t}\mathbf{z}_t + \frac{\sqrt{\bar{\alpha}_{t-1}}\beta_t}{1-\bar{\alpha}_t}\hat{\mathbf{z}}, \mathbb{I}(t > 1)\beta_t\mathbf{I})$
9:     **end for**
10:     Compute future states for each latent $\mathbf{z}_0^i$: $\mathbf{s}_H^i = p_\eta(\mathbf{s}_H^i \mid \mathbf{s}_0, \mathbf{z}_0^i)$
11:     Find best skill based on the cost-function: $i = \operatorname{argmin} \ \mathcal{J}(\mathbf{s}_H^i)$
12:     $h = 0$
13:     **while** $h < H$ **and** not *done* **do**            ▷ Execute Skill $\mathbf{z}_0^i$
14:         Observe environment state $\mathbf{s}_h$
15:         Get action $\mathbf{a}_h \in \pi_\theta(\mathbf{a} | \mathbf{s}_h, \mathbf{z}_0^i)$
16:         Execute action $\mathbf{a}_h$
17:         Update *done*
18:         $h = h + 1$
19:     **end while**
20: **end while**

---

The cost-function which we use for the goal-reaching environments is the Euclidean distance to the goal. We can also extend this planning to horizons greater than $H$ by further sampling latents for each future state $\mathbf{s}_{t+H}^i$ ($1 \le i \le n$). This means, after sampling $n$ latents for $\mathbf{s}_t$ with the diffusion prior, we further sample $n$ more latents for each of the future states $\mathbf{s}_{t+H}^i$. This increases the 'planning depth' $d$. The final states at the last level of planning are then scored using the cost-function and the latent at the first level which led to that final state is chosen for execution. This procedure complexity grows exponentially and thus the planning depth has to be restricted. For a planning depth of $d$, there are $n^d$ model calls. We found a planning-depth of $d = 2$ to be sufficient for all navigation environments achieving state-of-the-art results. Thus, with a latent sequence horizon of $H = 30$, our total planning horizon is 60. The results are tabulated in Table 6.

Table 6: Performance comparison on D4RL navigation tasks with LDCP.

| Dataset | BC | BCQ | CQL | IQL | DT | Diffuser | DD | LDCQ (Ours) | LDGC (Ours) | LDCP (Ours) |
|---|---|---|---|---|---|---|---|---|---|---|
| maze2d-large-v1 | 5.0 | 6.2 | 12.5 | 58.6 | 18.1 | 123.0 | - | **150.1** ± 2.9 | **206.8** ± 3.1 | **184.3** ± 3.8 |
| antmaze-medium-diverse-v2 | 0.0 | 0.0 | 53.7 | 70.0 | 0.0 | 45.5 | 24.6 | **68.9** ± 0.7 | **75.6** ± 0.9 | **77.0** ± 1.1 |
| antmaze-large-diverse-v2 | 0.0 | 2.2 | 14.9 | 47.5 | 0.0 | 22.0 | 7.5 | **57.7** ± 1.8 | **73.6** ± 1.3 | **59.7** ± 1.3 |

### E.1 VISUALIZING MODEL PREDICTIONS

Learning a world model also allows us to visualize the effect of executing any given latent behavior. This means, even when the model is not used for planning, like in LDCQ, it can be used to compute the final state that will be reached for every latent behavior from a particular state. This information can be used to understand if the model is learning reasonable behavior modes.

We plot the *xy*-coordinates of our abstract world model $p_\eta(\mathbf{s}_{t+H} \mid \mathbf{s}_t, \mathbf{z})$ predictions at a $T$-intersection of AntMaze large environment for latents sampled from our diffusion prior $\mathbf{z} \sim p_\psi(\mathbf{z} \mid \mathbf{s}_t)$ in Figure 5 to demonstrate this. The plot shows that the diffusion prior sampled latents which go in all the three directions at the T-intersection.

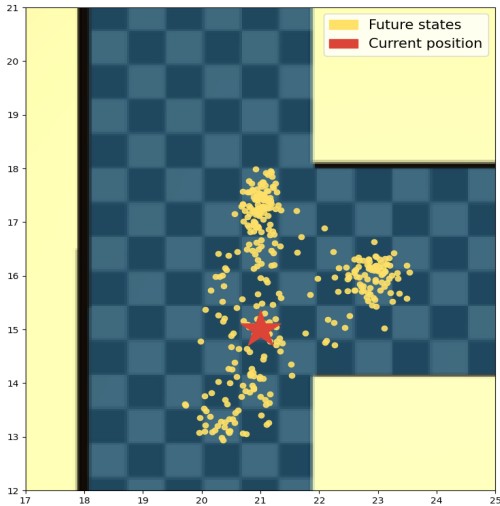

Figure 5: **Visualizing model predictions:** Visualization of future states with latents sampled from the diffusion prior at a T-intersection in antmaze-large-diverse-v2 D4RL task. We can see multimodal future state predidctions corresponding to 3 possible directions at the T-intersection.

# F   D4RL TASKS

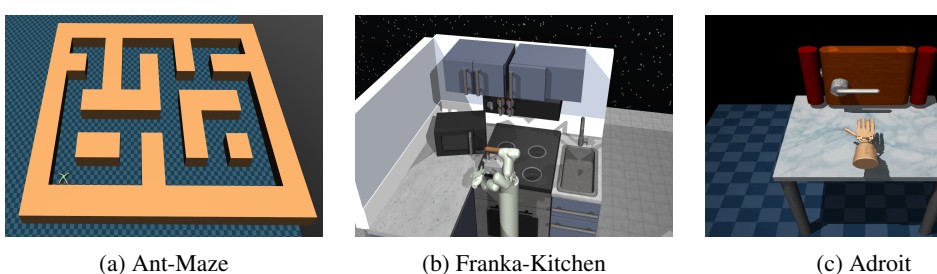

|(a) Ant-Maze|(b) Franka-Kitchen|(c) Adroit|

Figure 6: D4RL environments

The tasks of AntMaze and Franka Kitchen require long horizon credit assignment and stitching trajectories in the behavior dataset. We highlight consistently strong performance in these tasks as the primary empirical result of our method, since most other baselines perform poorly here, even other diffusion based methods. The Adroit suite also consists of narrow, but precision oriented manipulation tasks with sparse rewards. We also show very strong performance here. This task does not require trajectory stitching, but needs filtering of low reward demonstrations while sticking close to the behavior support.

The locomotion datasets are collected from SAC agents of varying performance. Our method has average performance on the locomotion suite while being much stronger in the other tasks. We suspect the high periodicity of the walking gaits in the locomotion suite does not benefit much from reasoning with temporal abstraction. We also do not use a perturbation function during Q-learning like Fujimoto et al. (2019), which makes it difficult for us to improve over the poor controllers in medium and medium-replay locomotion datasets. Introducing a perturbation function requires careful tuning to avoid extrapolation error, and the converged Q-learning wouldn't necessarily correspond to a high value policy, which is why other offline RL methods, which try to balance this tradeoff, evaluate online during training and consider the best scores. We however only evaluate a policy once after training is fully complete.

## G   CARLA AUTONOMOUS DRIVING TASK

To extend our method for tasks with high-dimensional image input spaces, we propose to compress the image space such that our method operates on this compressed state space. We create a low-dimensional compressed representation using an autoencoder $\mathcal{E}$ before using the LDCQ framework. Note that this encoder operates on a single image and not on a temporal sequence of images (Figure 7). The downstream framework of LDCQ however operates on the temporal compressed image sequences.

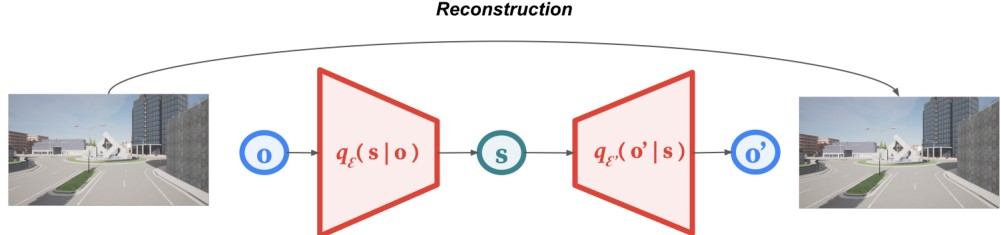

Figure 7: Autoencoder training for image-based task

We evaluate the performance of our method on the CARLA autonomous driving D4RL task. The task consists of an agent which has control to the throttle (gas pedal), the steering, and the break pedal for the car. It receives $48 \times 48$ RGB images from the driver's perspective as observations. We use a $\beta$-VAE architecture to create a 32-dimensional compressed state for this task. The horizon for LDCQ is set to $H = 30$.

## H   RANDOM WALK 1D

In this experiment, we construct a simple toy problem to show how sampling effectively from the multimodal behavioral distribution helps the diffusion prior outperform a Gaussian VAE prior during Q-learning. We construct a simple toy MDP with a one-dimensional state space $\mathcal{S} \in [-10.0, 10.0]$. The agent starts at the origin (0,0) and receives a reward of 10 if it reaches either the far left (-10.0) or far right (10.0) state, and -1 reward every other step. The environment times out after 500 steps. The action is the distance moved in that timestep with a max distance of length 1, $\mathcal{A} \in [-1.0, 1.0]$. The dataset consists of episodes where the agent randomly selects actions from the uniform distribution $a \sim \mathcal{U}([-1.0, -0.8] \cup [0.8, 1])$. This means the agent has a step size between 0.8 to 1.0 units either left or right every timestep. We train a VAE to try to fit this action distribution, and use BCQ to learn a policy. We also train train a diffusion based policy with LDCQ, using $H = 1$ and compare the results.

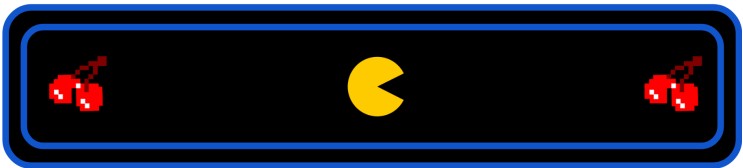

Figure 8: 1D Random walk

We find that the VAE frequently samples actions never present in the dataset. This is because the Gaussian mean to the above action distribution is 0.0, but no actual actions are present between $(-0.8, 0.8)$ where a large proportion of probability mass is assigned by the Gaussian model. Meanwhile, the diffusion prior is able to fit the 2 modes quite well. After 10000 steps of Q-learning, the diffusion constrained policy learns to navigate to either end perfectly and achieves an average reward of **-2.2** while the VAE constrained policy is still almost random, frequently taking actions with small step size and an average reward of **-66**.

## I  INCREASING DIFFUSION STEPS IMPROVES PERFORMANCE

We study the impact of the number of diffusion steps on the performance for LDCQ. We found that for the locomotion tasks, increase in diffusion timesteps $T$ during evaluation generally corresponds to increase in task performance. We plot these results in Figure 9.

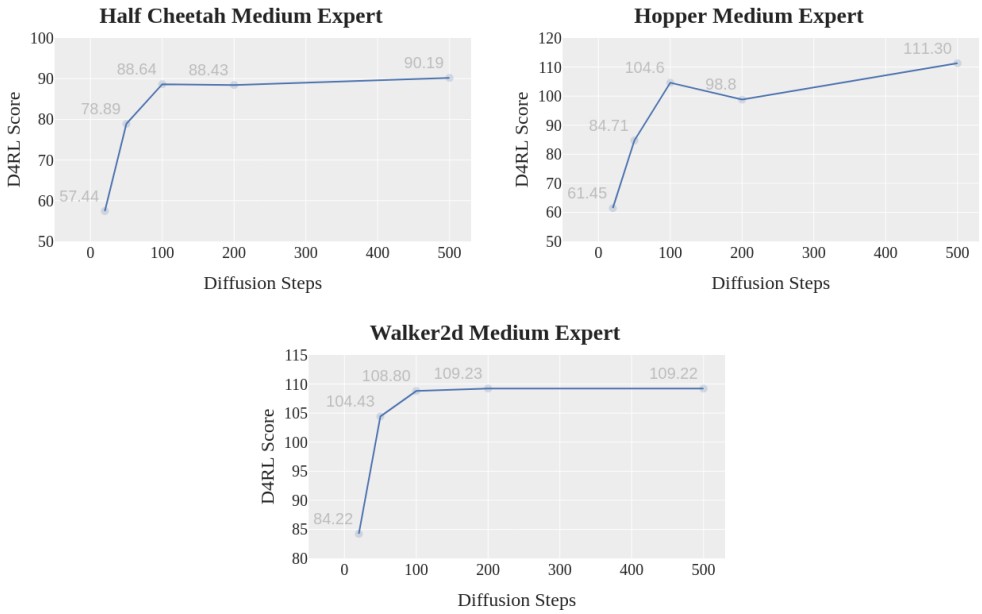

Figure 9: D4RL score for LDCQ with varying diffusion steps $T$ in locomotion tasks.

For the long horizon tasks, we found that increasing diffusion steps resulted in an initial trend upward in performance. Beyond this, the performance does not improve with additional diffusion steps (Figure 10).

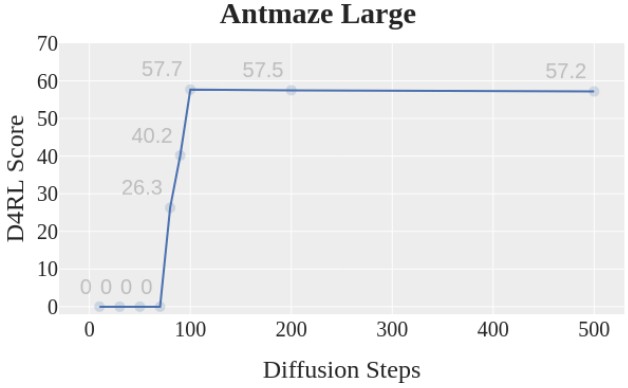

Figure 10: D4RL score for LDCQ with varying diffusion steps $T$

We also used additional diffusion steps at time $t = 0$ similar to Diffusion-X (Pearce et al. (2023)). This means that after the DDPM sampling of diffusion from time $T$ to 1, we run $X$ additional diffusion steps to further denoise the sample, assuming time-step $t = 1$. Pearce et al. (2023) reasoned that this pushes the samples further towards higher-likelihood regions. We used 10 additional steps across experiments and found this to slightly improve performance.

## J    PERFORMANCE IMPROVEMENT WITH TEMPORAL ABSTRACTION

We provided empirical evidence for improvement in performance as we increase temporal abstraction or horizon $H$ for the *kitchen-mixed-v0* environment. We see similar trends for the other long-horizon tasks as well (Figure 11). The performance in general improves with increasing temporal abstraction but beyond a certain point, it drops possibly because of the limited capacity of the policy decoder.

For the locomotion tasks, we did not observe any noticeable difference with increase in temporal abstraction, so we ended up using a moderate sequence length $H = 10$. This could be due to the high frequency periodicity of these tasks that does not require much look-ahead.

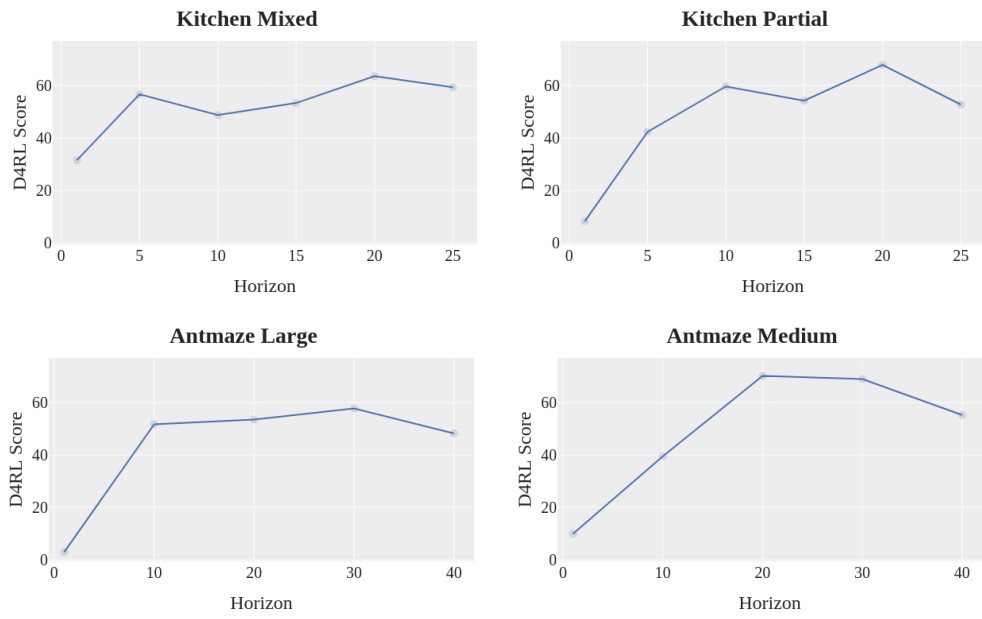

Figure 11: D4RL score for LDCQ with varying sequence horizons $H$.

## K    Q LEARNING TD ERRORS

We found Q learning with these skill latents to be very stable. This, alongside strong empirical performance in these tasks indicates that the skill latents learnt with the $\beta$-VAE contain useful information, and that the Q function is able to extract this information easily. We show the training graphs for antmaze-large, kitchen-partial, halfcheetah-medium, and Adroit door-human below. Each X-axis tick corresponds 3000 updates from minibatches sampled from the PER buffer. The Y-axis is the average TD error for those 3000 minibatches.

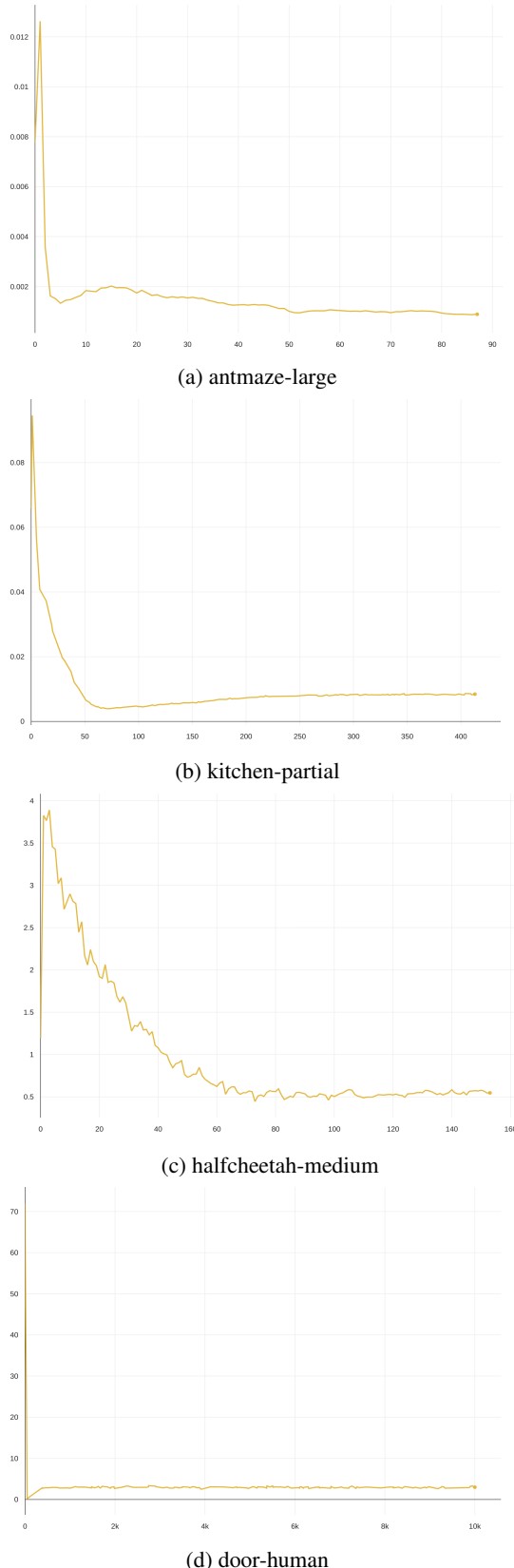

Figure 12: Q learning curves for different tasks. X axis is number of epochs of size 3000 from PER buffer, Y axis is average TD error over last 3000 minibatches.

