# OpenReview forum: "Reasoning with Latent Diffusion in Offline Reinforcement Learning"
_ICLR.cc/2024/Conference — ICLR 2024 poster_

### Official Review · Reviewer_LeBC · 2023-10-30

**Soundness:** 3 good
**Presentation:** 2 fair
**Contribution:** 3 good
**Rating:** 5
**Confidence:** 4

**Summary:**

This paper proposes using latent diffusion in offline RL as a way to compress task specific skills and information that helps learning Q-functions for improved performance in offline benchmarks. This paper provides a good instantiation for how to use latent diffusion models (LDM) in the offline RL setting. The key is to use a separate diffusion model, from which to learn a latent conditioned policy (similar to prior works using latent skills for RL). The proposed approach therefore seems quite useful for long horizon credit assignment tasks, where temporal abstraction and skills play a key role, and the work tries to justifies this through sparse reward environments for experimental evaluation.

**Strengths:**

This paper proposes an interesting idea that allows decoupling the diffusion model from the policy decoder; allowed the algorithm to be used for both continuous or discrete action environments. Experiment results are primarily for continuous action D4RL offline tasks, and shows relatively improved performance across few benchmarks. The execution of the algorithm is interesting, but I worry about the easy-ness of the approach. I agree with the authors that the reason that the algorithm is structured this way is to have better flexibility in terms of the latent skills from the diffusion prior, but see my comments in weaknesses. Other than that, the execution of the algorithm in the BCQ setting taking everything in the latent space (assuming the latents are good) is an interesting idea. Figure 2 shows how well the latents also get distributed across the horizons, suggesting that diverse skills can be learnt across horizons.

**Weaknesses:**

1. Experiment results are difficult to follow. The D4RL results are the primary ones, but the appendix claims to have results for CARLA and Goal Conditioning tasks too? It seems the goal conditioned tasks are not the standard ones in GCRL literature, and it is not clear what the key takeaway is other than the constrained offline RL results + qualitative results evaluating the latents across the horizon.

2. The proposed algorithm is interesting, but might have difficulty with the execution and easy-ness of the approach. Primarily because the Q-function is now learnt over Q(s,z) latents, and it would be helpful to do some qualitative evaluation of the Q-values over the latents. This is because the learnt Q would now heavily depend on the quality of the latents, which is coming from a beta-vae, and prior works on representation learning has showed that VAEs are not necessarily good for learning good quality latents.

3. This draws to my other worry of how good the latents are when learnt from Beta-VAE, and if there is a qualitative analysis about the structure that Beta-VAE latents recover from these tasks. Recent works have shown that with representation learning objectives, the latent structure can be well recovered - this work only shows how the latents are distributed across the horizonz, but it is not clear if these latents can itself contain useful information specific to the task?

4. Maybe not so related, but I wonder if an entropy-regularized objective can also be used in these settings to induce diverse skills learnt from the diffusion prior? The paper uses BCQ in offline, which is drawn from TD3 - so I agree that the algorithm is fundamentally using a deterministic policy approach; but for the discrete action tasks - maybe entropy regularized objectives to learn diverse skills may be introduced?

5. Paper claims that the algorithm can work for both continuous and discrete action spaces - but there aren’t enough results showing for discrete actions? Why is that so?

6. How does the claim on learning multi-modality for the diffusion approach compare with other representation objectives, rather than just the VAE? I think more baselines should be compared here, and I doubt that well tuned representation objectives would also be able to capture the multi-modality aspect similar to the diffusion model.

7. Experimental results should be better organized; appendix seems to contain more domains, but they are not referred to the main; there are figures for the environments, e.g CARLA, but I couldnt gind results for that in the paper?

**Questions:**

See my questions along with the weakness section. Primarily, my worries are :

1. How does the representations learnt with beta-vae compare with other representation objectives for learning the latents? How does diffusion compare to that?

2. How can we evaluate the quality of the latents? This is of big importance for this work, since the Q function is now learnt over the latents; I would like to see quantitivate results showing the effectiveness of the Q-function, maybe TD errors too, and qualitative results showing how good are the latents. Figure 2 only shows that latents are diverse and more skills are introduced across the horizons; but I worry about the use of Beta-VAE in the algorithm, and think better representation objectives should instead be used in the algorithm pipeline. Results justifying this would strengthen the paper.

---

> ### Author Response · Authors · 2023-11-16
> **Author response to review**
>
> Thank you for your questions. We have merged answers to some of them as they are related, and have attempted to address all of them:
>
> > ___Experiment results are difficult to follow.... It seems the goal conditioned tasks are not the standard ones in GCRL literature, and it is not clear what the key takeaway is other than the constrained offline RL results + qualitative result___
>
> The takeaway of D4RL CARLA experiment on carla-lane-v0 was to show that LDCQ scales more graciously than prior offline RL algorithms to image-based environments (last row of Table 1). The takeaway of GCRL experiment was to make a fair comparison with the Diffuser baseline, which uses goal-conditioned inpainting strategy. Our goal-conditioned latent diffusion method (LDGC) outperforms Diffuser by 65% on maze2D-large (Table 1, last column).
>
> >  ___I would like to see quantitivate results showing the effectiveness of the Q-function, maybe TD errors too, and qualitative results showing how good are the latents___
>
> Thanks for the insightful question. The $\beta$-VAE latents are trained to distill important information from the trajectory sequence. The usefulness of the latents is evidenced by the fact that the policy conditioned on the latents achieves significantly better performance than baselines that use regularized latents, such as OPAL [1].
>
> Q learning is only performed with the high-level latents, and the low-level policy is kept fixed after initial VAE training. If the latent representation does not contain information that is useful, then the low-level policy decoder would collapse to behavior cloning of the dataset, and so Q learning with the high-level policy will not affect it at all and our performance would be similar to BC baseline. ***We have a added TD error plot during Q learning in Appendix K Figure 12 for AntMaze, Franka, Adroit, and Locomotion tasks.*** This plot shows that the td error steadily decreases, indicating stable convergence of the Q-learning.
>
> > ___…but I worry about the use of Beta-VAE in the algorithm, and think better representation objectives should instead be used in the algorithm pipeline. Results justifying this would strengthen the paper.___
>
> Many of your questions seem to center around whether using a VAE leads to good representations. Vanilla VAEs are not necessarily the best forms of representation learning, many previous skill-learning papers use these for simplicity as well. Some of these papers were referenced by reviewer Agic (OPAL[1], SPiRL[2], PLAS[3]).
>
> We use $\beta$-VAE with a low value of $\beta$, which is a common choice for training latent diffusion models for other modalities, such as high-resolution images. The low value of $\beta$ ensures that the VAE behaves like an auto-encoder that encodes compressed latents with high mutual information with the uncompressed high-dimensional inputs. As a result, we learn more information-rich latents than these prior works by lowering the KL regularization and sampling from the resulting non-Gaussian latent space using the LDM prior. As such, we do not use the VAE itself as a generative model but just as a way to learn the skill-dependent low-level policy and get an information-rich latent space to be sampled by the LDM. The ability for this LDM to capture the diverse multimodal skill support in the behavior dataset is what allows us to use BCQ, as discussed in sections 5.1-5.3.
>
> > ___Maybe not so related, but I wonder if an entropy-regularized objective can also be used in these settings to induce diverse skills learnt from the diffusion prior?...___
>
> This could be an exciting future research direction that is beyond the scope of our current work. It is unclear to us what form of entropy regularized objective would be directly applicable to the current approach, but it could be interesting to force diversity in learnt skills. Would this make more sense in online RL where skill discovery is important than offline RL, where we just require copying skills from the behavior support?
>
> > ___Paper claims that the algorithm can work for both continuous and discrete action spaces - but there aren’t enough results showing for discrete actions? Why is that so?___
>
> We have not run experiments on discrete tasks because D4RL benchmark does not contain any such environment. We expect latent diffusion will work on discrete tasks as demonstrated in prior work [4].
>
> We hope we have sufficiently answered your questions. Please let us know if you would like any further clarification, or if you think we have not addressed any particular point..
>
> ***References follow in the next comment***

---

> > ### Author Response · Authors · 2023-11-16
> > **References**
> >
> > [1] Ajay, Anurag, et al. "Opal: Offline primitive discovery for accelerating offline reinforcement learning." arXiv preprint arXiv:2010.13611 (2020).
> >
> > [2] Pertsch, Karl, Youngwoon Lee, and Joseph Lim. "Accelerating reinforcement learning with learned skill priors." Conference on robot learning. PMLR, 2021.
> >
> > [3] Zhou, Wenxuan, Sujay Bajracharya, and David Held. "Plas: Latent action space for offline reinforcement learning." Conference on Robot Learning. PMLR, 2021.
> >
> > [4] Lovelace, Justin and Kishore, Varsha and Wan, Chao and Shekhtman, Eliot and Weinberger, Kilian. “Latent Diffusion for Language Generation.” NeurIPS, 2023

---

> ### Author Response · Authors · 2023-11-20
> **Kind request to reviewer**
>
> Dear Reviewer,
>
> We hope that you've had a chance to read our responses and clarification. As the end of the discussion period is approaching, we would greatly appreciate it if you could confirm that our updates have addressed your concerns.

---

> ### Author Response · Authors · 2023-11-22
> **Kind request for update**
>
> Dear Reviewer,
>
> We are rapidly approaching the end of the discussion phase. We hope we have answered your clarifying questions. We have made revisions to the paper to include the remaining D4RL tasks in Table 1. Our novel approach leveraging latent diffusion for TD learning using BCQ to train high level policies outperforms other offline RL methods in a majority of these tasks (maze2d, antmaze, franka, adroit and carla). We have also included further comparisons against other latent skill methods in Table 5, Appendix D. We would appreciate it greatly if the reviewer can acknowledge that we have addressed their concerns.

---

> ### Author Response · Authors · 2023-11-23
>
> Dear Reviewer,
>
> As the discussion phase is about to end, we would like to again kindly request if you could acknowledge the updates we have made to the paper and our response to your questions, and whether it has addressed your concerns.
>
> We believe most of your concerns stem from the usage of the beta-VAE, the choice which we have justified in our earlier response. We have also added TD error plots during training as per your suggestion as a preliminary analysis to the stability of Q learning with beta-VAE latents. We found training to be stable and reproducible, and the strong results in a majority of the D4RL benchmark suggest that the latents are information rich and that this information is easily extracted by the high level Q function during Q-learning. We acknowledge that there are other representation learning methods which can yield high quality representations, but our work instead focuses on sampling these weakly regularized information rich representations with latent diffusion (similar to image latent diffusion generative models), and how these skill priors can be used to learn high value policies in offline RL.
>
> We thank you for your feedback

---

### Official Review · Reviewer_hYz3 · 2023-10-31

**Soundness:** 4 excellent
**Presentation:** 3 good
**Contribution:** 3 good
**Rating:** 8
**Confidence:** 5

**Summary:**

# Summary

This work brings together the ideas from Batch Constrained Q-Learning (BCQ) and Latent Diffusion Models (LDMs), in order to succeed in offline RL. As in LDMs, the proposed method (called LDCQ - Latent Diffusion Constrained Q-learning) has a two stage training process. (i) The first stage trains a variational autoencoder (VAE) for trajectories (sequences of state-actions), where the decoder is the policy network. This latent space is a compressed representation of trajectories, serving as a form of temporal abstraction. (ii) The second stage of the training learns an LDM on this latent space and a latent-conditioned Q-network. At inference time, multiple latents are sampled from the LDM, and the Q-function is used to score each latent based on estimated Q-value. This scoring mechanism can be replaced with any kind of goal-conditioning (e.g.: L2 distance to goal in a maze environment), to obtain a goal-conditioned variant of the proposed method.

The use of latent diffusion makes this work distinct from recent diffusion-based offline RL methods such as Diffuser and Decision Diffuser. The analysis presented in this work highlights that (i) diffusion models are superior to VAEs when dealing with multi-modal latent spaces, (ii) the proposed LDCQ method achieves state-of-the-art or similar performance when compared with prior works.

**Strengths:**

# Strengths

The paper proposes a novel idea: Combining BCQ and latent diffusion models can lead to an offline RL algorithm that has the best of both worlds. Overall, this idea and the execution and analysis presented in this work is significant and of high quality.

1. The proposed method aims to fill a gap in existing methods such as Diffuser and Decision Diffuser by taking inspiration from latent diffusion models (LDMs): shifting diffusion into the latent space and separating the training process into policy and trajectory auto-encoding on one side and Q-learning and diffusion on the other side. With this choice, **the method inherits the strengths of both LDMs and batch-constrained Q-learning (BCQ)**.

1. The analysis of how the diffusion model of the proposed method (LDCQ) deviates from a baseline (that uses VAEs instead), clearly supports the hypothesis that **diffusion models are better at dealing with multi-modality** in the distribution of latents.

1. The paper has shown **similar or state-of-the-art performance** on a challenging selection of environments from the D4RL benchmark, the most notable of which is the performance on Franka kitchen-partial-v0 and kitchen-mixed-v0 environments, and the Ant-Maze large-diverse and medium-diverse environments. Specifically, two of these environments -- antmaze-large-diverse-v2 and kitchen-partial-v0 show significant improvements in the state-of-the-art.

1. The source code was provided.

**Clarity**: The paper has some issues with ordering of the introduced concepts and terminology, but by the end of the paper, the reader is left with a clear understanding of the presented work. The figures and algorithms are clear and greatly help in the understanding of the proposed method and analysis.

**Weaknesses:**

# Weaknesses

Overall, the paper has some statements/claims that are stretched, the empirical evidence seems like a mixed bag with certain D4RL environments/tasks silently omitted, and some weaknesses inherited from choosing latent diffusion models.

## Stretched claims
1. The phrases “improves credit assignment”, “faster reward propagation” describing the proposed work should be avoided, or backed by empirical evidence. I don’t see how either of these quantities can be measured. I understand how the proposed method will intuitively achieve both of these -- however, they are just intuitions, not proven.

1. The paper describes the decoder or policy as a “powerful decoder” that can handle “high-frequency details”. This is true for LDMs used for computer vision dealing with high resolution images. However, I don’t see how this translates well into RL. Training the VAE encoder-decoder model in a separate first stage seems like it will produce a rigid, fixed policy (i.e. decoder). The paper itself acknowledges that the “limited decoder capacity” is probably what is leading to the saturation of performance in Figure 4. How can the decoder be “powerful” and “limited” at the same time? Is there a way to empirically measure the strength of a decoder?

## Issues with empirical evidence

1. In Table 1, not all environments/tasks from D4RL are included. For example, the medium and umaze versions of Maze-2D are missing. In my experience, despite being easier environments, performance on these environments do highlight subtle differences between algorithms. They are certainly more important than the latter category of environments that do not require trajectory stitching.

1. In the latter category of environments that do not require trajectory stitching, more environments/tasks seem to have been silently omitted. The “-expert” suffix versions of Adroit pen, hammer and relocate tasks have been omitted. I would like to see an explanation of why certain environments in Table 1 were omitted, as otherwise it appears as cherry picking. Note that these results were not present in the Appendix either.

1. Some qualitative results such as visualization of trajectories at test time would have been helpful for supporting the trajectory stitching property.

1. Confidence intervals (error bars) are missing in Figure 4 and Adroit environment/tasks of Table 1.

1. Not all reported performance values are state-of-the-art, this claim should be qualified.

## Clarity

1. Batch-constrained Q (BCQ) is mentioned several times and the acronym is used once before it is explained in Section 4.2. I recommend briefly explaining BCQ as a part of “Section 3.2 Offline Reinforcement Learning”.

1. Algorithms 1 and 2 explain the second stage of training and the policy execution phase respectively. However, there is no algorithm box for the first stage of training (VAE). The policy \pi also has a subscript \theta, but \theta is not used anywhere else in the paper. For completeness, I would recommend adding an Algorithm to describe the first stage of training where the policy parameters \theta are learned.

-----------------
## Updated score after author response

The authors have addressed the major concerns raised above and I have increased my rating from 6 to 8.

**Questions:**

# Questions

Some of my questions have been merged into the weaknesses section. Here are the remaining questions.

- The policy execution first samples a latent z-i (representing a skill), and then fixes that z sample throughout the policy execution for H steps. What would happen if z-i is re-sampled after every action is taken by the policy (like model predictive control - MPC). Do you expect it to improve performance and if so, will it be much more computationally expensive given the cost of sampling z?

---

> ### Author Response · Authors · 2023-11-16
> **Author response to review (Part 1 out of 2)**
>
> Thanks for your detailed review. We have tried to address most of your points in our revised draft, and have provided some more context below:
>
> Stretched claims:
> -----------------------
>
>  >  ___The phrases “improves credit assignment”, “faster reward propagation” describing the proposed work should be avoided, or backed by empirical evidence.___
>
> We agree these are more intuitive claims rather than something which we quantitatively measured. Qualitatively, improved performance in long horizon tasks indicates better credit assignment. We can change the wording in the abstract to reflect this for the final camera ready version, since abstract changes are not permitted during the review phase.
>
>  > ___The paper describes the decoder or policy as a “powerful decoder” that can handle “high-frequency details”___
>
> We intended to refer to our decoder as “powerful” in comparison to other raw trajectory diffusion methods. Due to the separation of the low-level policy from the high-level latent diffusion prior, we can have more powerful decoder networks, which would be infeasible to train with end-to-end diffusion methods. We use an autoregressive decoder, which outputs the action dimensions one at a time sequentially. It is possible for us to use even larger decoders, but we chose not to do so because it would require extra computational resources and be time inefficient for a marginal gain in performance, and it can also result in overfitting because D4RL datasets are not very large or diverse.
>
> Empirical evidence:
> -------------------------
>
>  >  ___In Table 1, not all environments/tasks from D4RL are included. For example, the medium and umaze versions of Maze-2D are missing.___
>
> Thanks for pointing this out. ***We have added these experiments and antmaze-umaze-diverse to Table 1.*** Our proposed LDCQ methods  outperforms other baselines in the smaller mazes as well. We agree that they can still be useful results alongside the harder antmaze tasks.
>
> >  ___The “-expert” suffix versions of Adroit pen, hammer and relocate tasks have been omitted.___
>
> ***We have added the Adroit ‘expert’ tasks as well as ‘kitchen-complete’ to Table 1.*** We did not cherry-pick D4RL tasks. We only had a limited compute budget, so the reasoning behind not including these previously was that the datasets for these tasks only consist of expert demonstrations and are better for testing behavior cloning ability rather than offline RL. Further, existing state-of-the-art benchmarks IQL[1] and CQL[2] also did not evaluate Adroit expert tasks in their papers. However, we agree that more task evaluations can highlight the robustness of the algorithm better, and we have added the experiments to Table 1. We found our method performed very well in these tasks compared to baselines, possibly because of the improved behavior cloning capabilities with diffusion models. We summarize the results here:
> |                       | BC    | BCQ   | CQL   | IQL  | DT   | Diffuser | DD | LDCQ       | LDGC       |
> |-----------------------|-------|-------|-------|------|------|----------|----|------------|------------|
> | maze2d-umaze          | 3.8   | 12.8  | 5.7   | 47.4 | 27.3 | 113.5    | -  | 134.2+-4.0 | 141.0+-2.7 |
> | maze2d-medium         | 30.3  | 8.3   | 5.0   | 34.9 | 32.1 | 121.5    | -  | 125.3+-2.5 | 139.9+-4.2 |
> | antmaze-umaze-diverse | 45.6  | 55.0  | 84.0  | 62.2 | 54.0 | -        | -  | 81.4+-3.5  | 85.6+-2.4  |
> | kitchen-complete      | 65.0  | 52.0  | 43.8  | 62.5 | -    | -        | -  | 62.5+-2.1  | -          |
> | pen-expert            | 85.1  | 114.9 | 107.0 | -    | -    | -        | -  | 121.2+-3.6 | -          |
> | hammer-expert         | 125.6 | 107.2 | 86.7  | -    | -    | -        | -  | 45.8+-10.5 | -          |
> | door-expert           | 34.9  | 99.0  | 101.5 | -    | -    | -        | -  | 105.1+-0.3 | -          |
> | relocate-expert       | 101.3 | 41.6  | 95.0  | -    | -    | -        | -  | 104.7+-1.4 |            |
>
>  > ___Some qualitative results such as visualization of trajectories at test time___
>
> Unlike Diffuser, we do not predict the full trajectory at once. Since we do Q learning, greedily predicting the next best H-length sequence of actions. As such, it is difficult to show the stitching ability that happens during diffusion since we only diffuse smaller sub-trajectories at a time. The complete trajectory for tasks like antmaze look similar for all algorithms, with the main failure mode mainly just toppling of the ant after hitting walls. Let us know if you had something else in mind, and we can try to create these visualizations.
>
> ***RESPONSE CONTINUED IN NEXT COMMENT***
>
> [1] Kostrikov, I., Nair, A., and Levine, S. Offline reinforcement learning with implicit Q-learning. In International Conference on Learning Representations, 2022.
>
> [2] Kumar, A., Zhou, A., Tucker, G., and Levine, S. Conservative Q-learning for offline reinforcement learning. In Advances in Neural Information Processing Systems, 2020.

---

> ### Author Response · Authors · 2023-11-16
> **Author response to review (Part 2 out of 2)**
>
> ***READ PART 1 FIRST***
>
> >  ___Confidence intervals (error bars) are missing in Figure 4 and Adroit environment/tasks of Table 1.___
>
> We have added the missing error bars in Adroit, over 5 runs of the algorithm.
>
>  >  ___Not all reported performance values are state-of-the-art___
>
> We have not claimed SOTA in the locomotion tasks, and in Appendix F we had provided potential reasons why our method seems to underperform in these tasks. To summarize, we suspect the high-frequency periodicity of the walking gaits does not require temporal abstraction, and our lack of addition of a perturbation function like in BCQ for simplicity of tuning means we do not trade off conservatism from the behavior support. As shown in Table 1, we do have very strong results in all non-Locomotion tasks compared to baselines.
> Clarity:
>
> >  ___Batch-constrained Q (BCQ) is mentioned several times and the acronym is used once before it is explained___
>
> We now reference BCQ when we introduce the idea of batch-constrained action selection in the related work section.
>
> >  ___I would recommend adding an Algorithm to describe the first stage of training___
>
> ***We have added the beta-VAE and diffusion prior training to Algorithm 2 in Appendix B, referenced in section 4.1.***
>
> Questions:
> --------------
>
> > ___What would happen if z-i is re-sampled after every action is taken by the policy___
>
> We found that planning skills at a higher frequency did not seem to affect performance in any measurable way in antmaze tasks, and improved slightly but within the margin of error for Franka tasks. As we show in Appendix I, increasing diffusion steps seems to meaningfully improve performance, and high-frequency high-level planning with more diffusion steps would be very inefficient/slow.
>
> We hope we have addressed your concerns. Please let us know if you would like us to clarify anything else.

---

> > ### Comment · Reviewer_hYz3 · 2023-11-22
> > **Acknowledgement of author response and change in score**
> >
> > Thank you for the response and addressing all of the major concerns that I had initially raised. The complete set of experiments and the clarification of what "power decoder" implies was very helpful.
> >
> > The paper is in better shape now and the strengths I had listed in my original review make this a great contribution. I will update my rating in the main review to increase my score from 6 to 8.

---

> ### Author Response · Authors · 2023-11-20
> **Kind request to reviewer**
>
> Dear Reviewer,
>
> We hope that you've had a chance to read our responses and clarification. As the end of the discussion period is approaching, we would greatly appreciate it if you could confirm that our updates have addressed your concerns.

---

> ### Author Response · Authors · 2023-11-22
> **Kind request for update**
>
> Dear Reviewer,
>
> We are rapidly approaching the end of the discussion phase. We have addressed the comments in the review, and have made revisions to the paper to include the remaining D4RL tasks in Table 1. Our novel approach leveraging latent diffusion for TD learning using BCQ to train high level policies outperforms other offline RL methods in a majority of these tasks (maze2d, antmaze, franka, adroit and carla). We have also included further comparisons against other latent skill methods in Table 5, Appendix D. We would appreciate it greatly if the reviewer can acknowledge that we have addressed their concerns.

---

### Official Review · Reviewer_Agic · 2023-11-01

**Soundness:** 3 good
**Presentation:** 3 good
**Contribution:** 2 fair
**Rating:** 6
**Confidence:** 4

**Summary:**

This paper proposes a novel offline RL algorithm that leverages a diffusion model to plan over the learned temporally abstract latent space for action representation. The empirical results show that temporal abstraction can help distinguish latent skills and the proposed method shares competitive performance with other state-of-the-art baselines.

**Strengths:**

Exploring how to leverage expressive models such as diffusion models and Transformers for policy learning is an important direction in RL, especially for the offline setting. To the best of my knowledge, the idea of combining high-level diffusion planning and low-level primitive learning is novel. The paper is well-organized and clearly written.

**Weaknesses:**

While well-motivated, I have some questions about this work:

1. My major concern is that the results on offline RL benchmarks may be insufficient to show the advantage of planning with diffusion models on latent action space. While I appreciate that the authors have covered the most popular state-of-the-art baselines in Table 1, I think it is necessary to compare LDCQ with some literature that similarly performs planning on the learned action representation space learned by VAE [1, 2, 3, 4] or Flow [5, 6]. Also, I am curious about the ablation of LDCQ on diffusion models (Section 5.2 has shown some support, however, it compares the diffusion prior with the VAE conditional prior rather than a learned latent policy).
2. The paper has discussed some limitations of prior works depending on VAE representations. However, it is unclear to me why introducing latent diffusion can solve these limitations, as LDCQ also requires a learned VAE embedding. Is it because the diffusion model is more expressive so you can use less KL penalty or there are some other reasons? And if we learn the latent action space with a flow model, which is a lossless representation, does latent diffusion planning still have some advantages?

Overall, while this paper presents some interesting ideas, I am unable to recommend acceptance at this stage given the questions mentioned above. However, I would be happy to raise my score if the authors could address my concerns.

[1] Pertsch, Karl, Youngwoon Lee, and Joseph Lim. "Accelerating reinforcement learning with learned skill priors." Conference on robot learning. PMLR, 2021.

[2] Ajay, Anurag, et al. "Opal: Offline primitive discovery for accelerating offline reinforcement learning." arXiv preprint arXiv:2010.13611 (2020).

[3] Zhou, Wenxuan, Sujay Bajracharya, and David Held. "Plas: Latent action space for offline reinforcement learning."  Conference on Robot Learning. PMLR, 2021.

[4] Chen, Xi, et al. "Lapo: Latent-variable advantage-weighted policy optimization for offline reinforcement learning." Advances in Neural Information Processing Systems 35 (2022): 36902-36913.

[5] Singh, Avi, et al. "Parrot: Data-driven behavioral priors for reinforcement learning." arXiv preprint arXiv:2011.10024 (2020).

[6] Yang, Yiqin, et al. "Flow to control: Offline reinforcement learning with lossless primitive discovery." Proceedings of the AAAI Conference on Artificial Intelligence. Vol. 37. No. 9. 2023.

**Questions:**

There are some questions and concerns, which I have outlined in the previous section.

UPDATE: I updated my score from 5 to 6 as the authors addressed most of my concerns in the response.

---

> ### Author Response · Authors · 2023-11-16
> **Author response to review**
>
> We thank the reviewer for sharing their concerns. We aim to address all concerns in the following response.
>
> > ___compare LDCQ with some literature that similarly performs planning on the learned action representation space learned by VAE or Flow___
>
> A1: Thanks for raising this interesting point. We have added a comparison with OPAL and PLAS among VAE methods, and Flow to Control among offline RL methods that use normalizing flows. For OPAL scores we used the code provided to us by the authors. SPiRL is an online RL method, and if used offline would resemble OPAL. All other remaining references suggested by the reviewer also do not have open-source codebases, making them hard to reproduce. We will add fair comparisons with the remaining references in the camera-ready version.
>
> LDCQ (ours) vastly outperforms OPAL and PLAS (VAE methods) across tasks. While Flow to Control reports better scores, we found it difficult to tune our implementation of their method. We have contacted the authors for their code, and have currently compared with the reported scores in their paper. ***We have only made comparisons in the tasks these papers quoted in their respective papers. Please refer to Appendix D, Table. 5 of the revised paper on OpenReview*** (and our global rebuttal). We list here the scores in antmaze and kitchen. The other scores are in the paper:
> |                        | OPAL | PLAS | Flow to Control | LDCQ      |
> |------------------------|------|------|-----------------|-----------|
> | antmaze-medium-diverse | 57.5 | 0.0  | 83.7            | 68.9+-0.7 |
> | antmaze-large-diverse  | 52   | 0.0  | 52.8            | 57.7+-1.8 |
> | kitchen-partial        | 55.5 | 43.9 | 74.9            | 67.9+-0.8 |
> | kitchen-mixed          | 50.2 | 40.8 | 69.2            | 62.3+-0.5 |
>
>
>
> > ___Is it because the diffusion model is more expressive so you can use less KL penalty or there are some other reasons?___
>
> A2: That is correct. The reason this works with a $\beta=0.05$ is because the diffusion model is expressive enough to capture the more unstructured but information-rich latent space produced with small $\beta$. This was not possible with the previous methods which learnt VAE embeddings and used VAE prior. See Figure. 3 for a PCA visualization of the diffusion latent space and the comparison with VAE prior.
>
>  > ___And if we learn the latent action space with a flow model, which is a lossless representation, does latent diffusion planning still have some advantages?___
>
> A3: We believe latent space planning compresses information across longer timescales into low-dimensional embeddings, which helps with computational/time efficiency of diffusion.
>
> However, if the latent space is lossless, and is an invertible function of the original state-action timeseries, then it is probably better to directly diffuse the raw time-series sequence. Moreover, there exists an asymmetry in the encoding and decoding stages, as we encode state-action sequences, but we only decode action sequences. While this can still be implemented via conditional NFs, it is simpler and more stable to directly diffuse the action sequence from states.
>
> In fact, diffusion models themselves are a particular type of continuous normalizing flow (CNF) through the probability flow ODE [1, 2]. Diffusion models are deep autoregressive normalizing flows that also conveniently have a fixed, tractable, simulation-free noising process, which allows us to train each reverse step independently. They are also more flexible with network architectures while normalizing flows require special architectural considerations for a diagonal Jacobian. As such, even if NFs might be sufficient for smaller tasks such as in D4RL and comparable in performance with latent diffusion, we claim that as we scale to large offline datasets, we expect diffusion models to be the choice of generative models that the community will turn to. Diffusion models have been shown to scale very well with large scale datasets in other generative domains. Our paper demonstrates a simple way to leverage these latent diffusion models for offline skill learning.
>
> [1] Yang Song, Jascha Sohl-Dickstein, Diederik P. Kingma, Abhishek Kumar, Stefano Ermon, and Ben Poole. Score-Based Generative Modeling through Stochastic Differential Equations. ICLR 2021
>
> [2] Lipman, Yaron, et al. "Flow Matching for Generative Modeling." The Eleventh International Conference on Learning Representations. 2022.
>
> Please let us know if there are any remaining questions. We look forward to continuing the discussion.

---

> > ### Comment · Reviewer_Agic · 2023-11-19
> >
> > Thank you for your clarification. I still have some questions understanding Figure 3. From this figure, I agree that under the same horizon, diffusion prior exhibits better capability in modeling multi-modal distribution for *training data* than *VAE prior*. However, I am wondering:
> >
> > 1. Does this result hold for trajectories during evluation time that are not exactly the same as training data?
> > 2. Does the trained VAE latent policy (as in OPAL) can capture the multi-modal distribution better?

---

> ### Author Response · Authors · 2023-11-20
> **Official Author Response to Reviewer Comment**
>
> We are happy to answer these questions.
>
> > Does this result hold for trajectories during evaluation time that are not exactly the same as training data?
>
> 1) Yes, this result holds during evaluation time as well. We show this in Section 5.3 (see Fig. 4), where we compare the D4RL evaluation performance of the same generative models shown in Fig. 3, i.e., latent diffusion (LDCQ) and VAE (BCQ-H). Both LDCQ and BCQ-H follow the same batch-constrained Q-learning (BCQ) setup to learn latent policies, with the only difference being the choice of the generative model to approximate the prior. This result suggests that diffusion models are not just better than VAEs at modeling the prior distribution in the latent space (as shown in Fig. 3), but they also generate policies that evaluate to a higher performance on the offline RL tasks (as shown in Fig. 4/Table 5).
>
> > Does the trained VAE latent policy (as in OPAL) can capture the multi-modal distribution better?
>
> 2) This is a great question. In theory, both VAE based methods such as OPAL and diffusion models with sufficient capacity can approximate any target distribution. However, it is commonly observed that diffusion models are better at handling multi-modal distributions than VAEs, as suggested by prior works in image generative modeling[1,2] and imitation learning[3]. Our improved results (shown in Table 5.) compared to OPAL suggest this is true in the offline RL setting as well.
>
> **Do these answers fully address the reviewer's concerns?** We look forward to continuing the discussion.
>
> References:
> ----------------
>
> [1] “Hierarchical Text-Conditional Image Generation with CLIP Latents”; Aditya Ramesh, Prafulla Dhariwal, Alex Nichol, Casey Chu, Mark Chen; https://arxiv.org/abs/2204.06125
>
> [2] “High-Resolution Image Synthesis with Latent Diffusion Models”; Robin Rombach, Andreas Blattmann, Dominik Lorenz, Patrick Esser, Björn Ommer; https://arxiv.org/abs/2112.10752
>
> [3] “Imitating Human Behaviour with Diffusion Models”; Tim Pearce, Tabish Rashid, Anssi Kanervisto, Dave Bignell, Mingfei Sun, Raluca Georgescu, Sergio Valcarcel Macua, Shan Zheng Tan, Ida Momennejad, Katja Hofmann, Sam Devlin; https://arxiv.org/abs/2301.10677

---

> > ### Comment · Reviewer_Agic · 2023-11-20
> >
> > Thanks for your further explanations. Your point 2 has not fully convinced me. While OPAL indeed learns a unimodal latent policy, it is conditioned on the state. Both Figure 2 and Figure 3 show that data points are clustering together according to the task so I wonder if a trained VAE latent policy is expressive enough for the latent action space. Section 5.3 as you mentioned in point 1 could support that diffusion latent policies are more flexible, and the comparison in Table 5 shows that LDCQ outperforms OPAL, however, we may need some more ablations (for instance, diffusion-based OPAL versus OPAL) to see the benefit is from the BCQ part or the diffusion part.
> >
> > But overall, I believe the authors' response has addressed most of my concerns, and I appreciate their additional results on comparing their method with VAE or Flow-based offline RL baselines and elaborating on the connection between diffusion models and flows. Table 5 shows that LDCQ can outperform its unimodal latent policy counterpart (OPAL and PLAS), and I feel that one possible improvement for this paper could be learning a Flow-based variant of LDCQ. Therefore I have increased my score to 6.

---

> > > ### Author Response · Authors · 2023-11-21
> > > **Official Author Response**
> > >
> > > We agree with the reviewer that further ablations can help highlight the subtle contributions of these components. We can work to add these to the camera ready version.
> > >
> > > We thank the reviewer for their valuable feedback!

---

### Author Response · Authors · 2023-11-16
**Global Response**

We have tried to address the reviewer’s concerns, by answering questions and through revisions to the paper adding new experiments and other information based on the feedback. We now summarize the major revisions to the paper:

1. ***We have added an extra Table 5 in Appendix D which compared our method with some other popular VAE/NF methods for offline RL***. Most offline RL algorithms are very sensitive to hyperparameter choices and we personally found it very hard to reproduce the results of several VAE/NF methods in the literature without open-source code. Also, we strongly believe that there needs to be simpler offline RL algorithms that work just as well with minimal changes on any benchmark tasks. This was a primary motivator for us to build an offline RL algorithm that is composed of simple building blocks with stable, easily reproducible training regimes. In particular, this is why we chose simple perturbation-free BCQ over CQL/Iusing an expressive autoregressive decoder. (Model architecture discussed in Appendix A.2).QL style loss so as to not require tuning conservatism, and diffusion over VAE/NF style generative models.

2. ***We added extra tasks from D4RL to Table 1***. These are the Maze2D UMaze and Medium, Franka kitchen-complete, Adroit expert suite. We match state-of-the-art baselines in all these tasks, with noteworthy improvement from baselines in the Adroit suite. We believe this is due to the more powerful latent Diffusion model being able to better learn the expert policy for the 24-DoF sensitive action space.

3. ***We have added an extra Algorithm description for the beta-VAE and diffusion prior training to Appendix B.***

4. ***We have added Q learning plots to Appendix K, Figure 12.*** These show that Q learning with the skill latents learnt using the $\beta$-VAE is smooth, addressing a concern from reviewer LeBC.

We also made some smaller edits to address other reviewer concerns, as indicated in our comments to them.

We highlight some of the main points the reviewers raised, and summarize our response:

Reviewer Agic:
--------------------

The reviewer wanted comparisons with related skill learning methods that use VAEs and Normalizing Flows. We have added this now as described earlier. The reviewer also wanted to clarify how our method would be better than latents from a VAE. We note that the weaker KL constraining makes the latents more information-rich, at the cost of making it harder to fit under a Gaussian prior. Further, the temporal abstraction with larger horizons also induces multimodality in the latent space as we show in Figure 2. Hence, latent diffusion allows us to gracefully cope with this information-rich, multimodal latent space and samples diverse skills from which we can do batch-constrained Q learning. We also argue that diffusion models are preferable to normalizing flows even if they are also expressive by being easier to train due to having simulation-free posteriors, less restrictive in architectures, and scaling well with large datasets.

Reviewer hYz3:
--------------------

The reviewer requested additional experiments on D4RL environments that were missing from our experiments. We have now added the results on these tasks in Table 1. We also tried to address some other concerns related to empirical and stretched claims.

Reviewer LeBC:
---------------------

Many of the reviewer’s questions revolve around the quality of the latent representations learnt by the beta-VAE. Following other latent diffusion methods, we instead use it much closer to an autoencoder that compresses a high-dimensional signal into a low-dimensional latent space, rather than a generative model like VAE. Making the latents information rich makes the trajectories more controllable with the skills. The added complexity in the latent space through weak regularization and temporal abstraction is handled gracefully by the expressive diffusion prior, which samples easily from the skills in the behavior support facilitating a strong BCQ based algorithm which shows very strong results across the D4RL benchmark suite. Other information-rich, and more disentangled representations can also be used from other representation learning methods as well with most of our proposed algorithm would be unchanged. However, we avoided this because we wanted our algorithmic components to be simple and focus on latent diffusion.

---

### Meta-Review · Area_Chair_PKEM · 2023-12-06

**Metareview:**

This paper introduces the idea of latent diffusion in offline RL. It trains a diffusion model on the trajectory latents from beta-VAE to get a better representation and then applies BCQ on top of the latents. It combines the benefit of diffusion model and BCQ. Experiments show the benefit against applying BCQ on the raw Beta-VAE latents and its better handling of multi-modal distributions.

There are some initial common concerns on the lack of offline RL experiments and comparison with other representation learning. The authors provide additional experiments from D4RL, and comparison with flow-based models. They addressed most concerns from reviewer Agic and hYz3.

The concerns of LeBC include
1. "easy-ness" of the execution, a lack of qualitative analysis of the structure of the learned latents

The authors' additional TD-error plot should partially alleviate the first concern.

2. lack of comparison with other representation learning methods other than VAE

Additional experiments include a comparison to flow-based methods on top of VAE latents. Comparison with other representation learning methods to get the raw latents is good to have but may not be necessary to justify the effectiveness of the propoesd usage of diffusion models.

3. lack of experiments on discrete actions.

The authors acknowledge this shortcoming. It would be useful to adjust the statement about its application to discrete action problems due to the lack of experimental evidence.

Given that most concerns have been alleviated during rebuttal and two reviewers raise their rating towards acceptance, I would recommend acceptance of this submission.

**Justification For Why Not Higher Score:**

Some weaknesses pointed out by review LeBC are valid.

**Justification For Why Not Lower Score:**

This work does show the benefit of learning a better representation from the raw beta-VAE latents. The combination of latent diffusion and BCQ in the latent space is interesting and novel.

---

### Decision · Program_Chairs · 2024-01-16

Accept (poster)